# Spatio-temporal evolution and distribution characteristics of the high-quality development of China's insurance industry

Heng Zhu[1], Bo Xiong[1], Meimei Tang[2]

1 School of Finance, Southwestern University of Finance and Economics, Chengdu, Sichuan, China,
2 School of Finance, Anhui University of Finance and Economics, Bengbu, Anhui, China

* zhuheng@swufe.edu.cn

## Abstract

This study, based on the criteria of "inclusiveness," "sustainability," and "internal-external structural coordination," establishes an evaluative framework for appraising the high-quality development of the insurance sector. It systematically gauges the overarching high-quality developmental status of China's insurance industry across regions. Employing non-parametric kernel density estimation, the Standard Deviation Ellipse, and spatial Markov chain, the investigation dynamically scrutinizes the national landscape of high-quality evolution within the insurance sector over the temporal spectrum. Furthermore, Moran's index and Dagum's Gini coefficient are harnessed to disentangle the spatial interdependence and heterogeneity characterizing the high-quality progression of the insurance industry among provinces. The findings disclose a pronounced regional development gap throughout China, surpassing intra-regional disparities and underscoring a notable concern of imbalance in regional insurance industry development. Despite the elevated development stature of the eastern region, substantial interprovincial differentials persist, exposing internal "inequities" within this region. In the central and western domains, although internal divergences in insurance industry development are gradually diminishing, the overall developmental benchmarks remain comparatively subdued.

## Introduction

The high-quality development of the insurance industry has become a global focus in the insurance market. The insurance industry plays a crucial role in economic growth, risk management, and social stability. Therefore, achieving high-quality development in the insurance industry is vital for safeguarding the interests of the economy and society. The impetus for the high-quality development of the insurance industry stems from various factors. The continuous development and expansion of the global economy bring about more risks and opportunities, necessitating more refined and innovative insurance products and services to meet evolving demands. The rapid development of technology and digital transformation has a significant impact on the insurance industry. The application of technologies such as artificial intelligence, big data analytics, and blockchain is changing the operational mode of insurance businesses and enhancing the customization and efficiency of insurance products.

**Data availability statement:** All relevant data are within the manuscript and its Supporting Information files.

**Funding:** This work was funded by the Southwestern University of Finance and Economics Research Interpretation Project on Building a Financial Power(Grant Number: JBKZD06002), awarded to ZH and XB. The funders had no role in study design, data collection and analysis, decision to publish, or preparation of the manuscript.

**Competing interests:** The authors have declared that no competing interests exist.

However, China's insurance industry still faces several challenges in achieving high-quality development. Fierce competition in the insurance market results in unreasonable insurance terms, affecting the industry's profitability and service quality. Incorrect risk assessment may lead to insurance companies assuming excessive risks, and the digitization and technological progress are altering the operational methods of the insurance industry, presenting a series of issues. In this context, clarifying the connotations of high-quality development in the insurance industry, analyzing the evolutionary characteristics of high-quality development, and laying the foundation for exploring the mechanisms and policy choices for the high-quality transformation of the insurance industry are essential.

High-quality development involves a comprehensive evaluation with multi-dimensional indicators. According to the classification of the measurement and evaluation subjects, it can be divided into macro-level [1], meso-level [2], and micro-level [3]. High-quality development in the insurance industry needs to construct an evaluation index from the nature of the industry itself, combined with the logical theoretical connotations of high-quality development. Currently, there is a lack of research on the construction of a comprehensive evaluation system for high-quality development in the insurance industry. Hou Xuhua [4] primarily measures the high-quality development level of internet insurance companies from five dimensions: insurance innovation, online operations, operational efficiency, insurance service economy, and risk control. Other scholars have studied aspects such as the inclusiveness of the insurance industry [5, 6], insurance efficiency [7,8], stability of the insurance industry [9], market structure of the insurance industry [10] and the relationship between the insurance industry and the digital economy [11].

Presently, several limitations exist in the research on the high-quality development of the insurance industry. Firstly, there is a notable absence of research focusing on the establishment of an evaluation index system for high-quality development within the insurance industry itself. While studies do exist regarding the high-quality development of specific insurance products like agricultural insurance [12–15], health insurance [16–18], and life insurance [19], there remains a dearth of research examining the overall high-quality development of the insurance industry comprehensively. Secondly, while some studies have touched upon aspects of the insurance industry's high-quality development [4,12,17,19–23], a significant gap exists in terms of theoretical research. Much of the existing research only offers explanations without robust grounding in relevant economic theories. Thirdly, inconsistencies in the statistical caliber of the number of insurance institutions across provinces over time lead to data distortion, impeding accurate analysis. Furthermore, the index system overlooks the inclusion of insurance intermediary institutions, which are vital components of the insurance market and significantly influence its healthy and stable development. Fourthly, there is a scarcity of research examining the interactive and dynamic changes of the insurance industry across time and space. While some studies have explored the distribution and evolution of the insurance industry concerning aspects like agricultural insurance [24,25] and social security [26], research into regional disparities in high-quality development across eastern, central, and western regions, as well as within these regions, remains limited. Lastly, existing research on the operational stability of the insurance industry primarily focuses on corporate entities, lacking relevant studies examining regional operational stability from a spatial perspective.

The paper makes significant contributions in several key areas. Firstly, it pioneers the construction of an evaluation index system for assessing the high-quality development of the insurance industry. This system is founded on three essential dimensions: "inclusiveness," "sustainability," and "internal-external structural coordination," originating from the intrinsic characteristics and challenges inherent to the insurance industry itself. Secondly, it employs the hp-filtering method to investigate the operational stability of regional insurance

industries, providing valuable insights into their performance dynamics. Thirdly, the paper offers a comprehensive time-dynamic analysis of the nationwide high-quality development of the insurance industry. This analysis encompasses the distribution of indices, spatial patterns, and developmental level transitions over time. Fourthly, the study employs Moran's index and Dagum's Gini coefficient to dissect the spatial correlation and heterogeneity characterizing the high-quality development of the insurance industry across various provinces. These methodologies enhance our understanding of the spatial dynamics and disparities within the insurance industry's development landscape.

## Theoretical connotations

Analyzing the Theoretical Connotations of "High-Quality Development" is the prerequisite and foundation for constructing the theoretical system and measurement indicators of "high-quality development" in the insurance industry. Currently, research on "high-quality development" generally starts from the perspective of the "new development philosophy" [2,27], grasping the characteristics of "innovation as the primary driving force, coordination as an endogenous feature, green as a universal form, openness as a necessary path, and sharing as the fundamental goal." It combines the research area with "high-quality development," constructs an indicator system, but lacks the core economic explanation of "high-quality development." The extension of "high-quality development" into specific research areas poses theoretical flaws. Therefore, this article attempts to start from the economic explanation of "high-quality development" and extend the theoretical connotations of "high-quality development" to the insurance industry, thus constructing a theoretical system and measurement indicators for the "high-quality development" of the insurance industry.

### Theoretical connotations of "high-quality development"

The economic explanation of "high-quality development" has two aspects: "development" and "quality." The "development theory" evolved from the "growth theory." In 1956, Solow built the neoclassical economic growth theory based on the Harrod-Doamr model established by Harrod [28] and Doamr [29]. Solow introduced the assumption of substitutability between capital and labor, suggesting that the economies of latecomer countries would converge to those of developed countries. However, Solow treated technological innovation as an exogenous factor, failing to explain the varying levels of technology among countries in reality. Therefore, Romer [30] incorporated knowledge and human capital into the production function to explain the differences in technology levels among different countries or regions, leading to different levels of economic growth. The research in the mid- to late 20th century analyzed the economic growth process from a single perspective through the use of mathematical models in economic abstraction. However, the multidimensional impacts of economic growth and related factors in reality are ignored. As the overall economic level and residents' income level increase, people's demands for living standards become more diversified. Some countries and scholars have attempted to incorporate non-economic factors related to economics, such as fairness [31], the environment, governance [1] into consideration. The study of "economic growth" has gradually shifted to "economic development." Therefore, "development" and "growth" have a general relationship. Commonly, both "development" and "growth" exhibit the characteristics of quantity and scale expansion. However, individually, the quantity and scale expansion of "growth" is in a single dimension. For example, "economic growth" is an expansion in the volume of the economy. Still, the quantity and scale expansion of "development" are multidimensional. For example, "economic development" not only requires an expansion of the economy's volume but also optimization in efficiency, structure, environment, etc.

The economic explanation of "quality" is lacking in the current mainstream Western economic theory system [27]. Western economic analysis systems choose to ignore the different characteristics of products to achieve the summation of products with different attributes into the production function for analysis. They assume that products have homogeneity and are uniformly presented using prices. Therefore, "quality" is abstracted, or "quality" is represented by prices. Simultaneously, a hypothesis is created that "quality" is equal to or positively correlated with prices. Thus, in the Western economic analysis system, when facing the phenomenon of "unrelated quality and price" or "high quality at a fair price," they can only choose to avoid it because of the lack of tools for the economic study of "quality" [32]. Therefore, the economic meaning of "quality" points to the acceptability of consumer demand for the use value of goods. The higher the "quality" of goods, the higher the level of use value that satisfies consumer demand, the easier the exchange, and the healthier the development of the market economy. However, it should be noted that consumer demand is changeable. According to Maslow's hierarchy of needs theory and Marx's theory of comprehensive development, human needs will continue to grow and evolve with the improvement of their own situation and the development of the economy or society. The economic meaning of "quality" points to the acceptability of consumer demand for the use value of goods. The higher the "quality" of goods, the higher the level of use value that satisfies consumer demand, the easier the exchange, and the healthier the development of the market economy. However, it should be noted that consumer demand is changeable. According to Maslow's hierarchy of needs theory and Marx's theory of comprehensive development, human needs will continue to grow and evolve with the improvement of their own situation and the development of the economy or society. The economic meaning of "quality" points to the acceptability of consumer demand for the use value of goods. The higher the "quality" of goods, the higher the level of use value that satisfies consumer demand, the easier the exchange, and the healthier the development of the market economy. However, it should be noted that consumer demand is changeable. According to Maslow's hierarchy of needs theory and Marx's theory of comprehensive development, human needs will continue to grow and evolve with the improvement of their own situation and the development of the economy or society. In summary, although the "quality" and "development" in "high-quality development" have different economic explanations, they are connected. "Development" involves multidimensional quantity and scale expansion, but this expansion is not disorderly. "Quality" is the orientation of this expansion, that is, human needs. Taking humans as the center is the orientation of this multidimensional quantity and expansion. If this "development" cannot meet the increasing demands of people for a better life, it is not "high quality."

## Theoretical connotations of high-quality development of the insurance industry

The "high-quality development of the insurance industry," as a form of "high-quality development" within a singular industry, requires an understanding of the industry's positioning in the economic and social context. First, if the industry deviates from its original positioning in the economic and social context, meaning it does not meet the increased demands of its trading partners, it cannot be considered "high-quality development." Second, since the products and services provided by the industry can target individual consumers or other independent businesses or industries, the resulting demands may differ, affecting the requirements for "high quality." In light of this, this paper attempts to categorize the recipients of insurance industry products and services into the economic and social ends. Because the demands of both exhibit characteristics of unity in commonality and individuality, the measurement

indicators for the high-quality development of the insurance industry are divided into three dimensions: "inclusiveness," "sustainability," and "internal and external structural coordination."

The "inclusiveness" dimension of the insurance industry measures the breadth and depth of its coverage. Currently, the contradiction between the increasing demand for insurance among the growing impoverished population and their limited payment capacity highlights the discrepancy between the inclusive practices of existing insurance policies for the impoverished and the targeted requirements for preferential treatment of the impoverished [33]. The concept of shared development emphasizes that "sharing is for everyone," and "shared development is the entitlement of all individuals, with each getting their fair share, not for a few or a part of the people." Therefore, the insurance industry cannot ignore the insurance needs of these vulnerable groups, namely the demand for fairness. On the economic side, attention needs to be given to the risk protection and financing needs of small and medium-sized private enterprises. This sector contributes significantly to tax revenue, domestic production value, technological innovation, urban labor employment, and the total number of enterprises. It is a crucial force in narrowing the wealth gap and achieving common prosperity. From the perspective of macroeconomics, a significant wealth gap may lead to overproduction if domestic production capacity exceeds domestic consumption capacity, necessitating the export of products or facing economic difficulties [34]. On the social side, low-income groups are the focal point of fairness concerns in the insurance industry. According to risk society theory, addressing the distribution of risks and alleviating harm are core issues in a risk society. Low-income families are more prone to falling back into poverty under the exposure of risks. Firstly, they rely mainly on the labor income of the family's breadwinner for income, making them vulnerable. Any impact on this single income source leads to a loss of income and a return to poverty. Secondly, due to the existence of risks and the lack of risk management tools, low-income families tend to save and reduce spending on education, hindering the improvement of human capital and access to higher income channels. The insurance industry, through risk sharing and transfer, provides risk management and compensation means for low-income families, enhancing the living standards of this group. Therefore, the requirement for the "inclusiveness" of China's high-quality insurance development involves expanding insurance coverage and reaching a larger population, so that as many groups as possible can benefit from insurance products and services. However, the current coverage breadth and depth of China's insurance industry still have significant room for improvement. In 2022, insurance penetration stood at 3.88%, the lowest level since 2016, while insurance density was 3,326 yuan, which is below the global average of 818 USD. The State Council's 2014 "Several Opinions on Accelerating the Modern Insurance Service Industry" set a target for 2022 of achieving an insurance penetration rate of 5% and an insurance density of 3,500 yuan. Thus, there remains a gap between the current level of insurance inclusiveness and the planned targets.

In 1987, the World Commission on Environment and Development defined "sustainable development" as development that "meets the needs of the present without compromising the ability of future generations to meet their own needs." Green finance theory suggests that the financial industry should avoid excessive speculation driven by short-term interests [35]. Sustainable development in the insurance industry requires meeting the current demand for insurance while not compromising the ability of future generations to meet their insurance needs [36]. On the business side, if the quality of insurance policies is not high, it not only fails to meet the needs of insurance consumers but also increases operating pressure for enterprises due to high maintenance costs. This weakens the future supply capacity of insurance products, unable to meet the insurance needs of future generations, indicating long-term demand [9]. On the

operational side, as a special industry in risk transactions, the risk management capability of the insurance industry directly affects its solvency when insurance incidents occur. The demand side of insurance products ultimately focuses on whether compensation can be obtained and the level of compensation when insurance incidents occur. Therefore, the risk management capability of the insurance industry is an important aspect of the "quality" of insurance products.

The concept of coordinated development emphasizes the overall, balanced, and coordinated nature of development. Whether the structure of the insurance industry is coordinated is determined by the structure of insurance demand and whether it can serve the real economy. With the development of a market economy and the improvement of people's living standards, the demand for insurance protection provided by insurance will evolve toward diversification and specialization, such as medical protection, innovation and entrepreneurship protection, and export risk protection. However, excessively focusing on asset-driven insurance business not only affects the financial stability of insurance companies [37] but also leads to homogenized competition with other financial industries like banks, overlooking its own risk protection function [38]. For example, in the property and casualty insurance sector, auto insurance premiums account for more than half of the total premium income, reaching 55.2% in 2022. Moreover, the auto insurance market remains dominated by a single major company, indicating a lack of competition and diversity in this segment. According to monopoly theory, suppliers in a monopolistic market demand high monopoly profits. Based on this, constructing a two-sector economic growth model, Shao [39] found that an overly concentrated insurance market demands excessive monopoly profits, leading to a decline in the production level of the real economy and impacting economic growth. The problems of adverse selection and moral hazard caused by information asymmetry in the insurance market may result in inefficient resource allocation and market inefficiency. An intermediary entity in the insurance market, acting as a bridge between policyholders and insurers, not only reduces information asymmetry and information collection costs but also promotes the completion of insurance market transactions. Moreover, it allows insurance companies to have more surplus funds for innovative product development, promoting the specialized operation of original insurance companies and enhancing the protection capabilities of insurance products. At the same time, the integration of digital technology with the insurance industry has given rise to a new sector known as "Insurtech." This development has significantly enhanced the production capabilities and overall productivity of the insurance industry, thereby altering its production structure. For instance, in 2020, digital technology enabled the insurance industry to achieve an industry average underwriting automation rate of 55.77%, an underwriting automation rate of 64.71%, and a claims automation rate of 21.48%. These advancements have opened up opportunities for innovative service and risk management solutions, providing a technological foundation for transforming the business output structure of the insurance industry.

In conclusion, based on the economic interpretations of "development" and "quality," this paper proposes a definition for "high-quality development" in the insurance industry. It is defined as the industry's development over a certain period, leading to changes in the satisfaction level of consumers' needs and values for insurance products and services. This definition also emphasizes the industry's ability for sustainable improvement over time. Both the economic and social aspects, serving as demanders for insurance products and services, contribute to the industry's requirements for fairness, long-term viability, obtainability of payments, multidimensional protection, and fulfilling the needs of economic growth. Therefore, the paper selects three dimensions—namely, "inclusiveness," "sustainability," and "internal-external structural coordination"—to measure high-quality development in the insurance industry. The three dimensions of high-quality development in the insurance industry are not in a parallel relationship; rather, they are based on the inclusiveness of the insurance industry. This foundation leads to new

requirements for the sustainability and internal-external structural coordination of the industry. Furthermore, the sustainability of the insurance industry provides stability for the expansion of inclusiveness, while the internal-external structural coordination sets target requirements for inclusiveness. The concept of "high-quality development" is not solely focused on outcomes but is a systematic approach that unifies processes and results. In this approach, inclusiveness serves as an outcome indicator, while sustainability and internal-external structural coordination serve as process indicators, aiming to achieve a unified perspective on the result and process requirements for high-quality development in the insurance industry (Fig 1).

## Measurement

### Index system

Based on the defined connotations of high-quality development in the insurance industry, this paper, starting from the three dimensions of "inclusiveness," "sustainability," and "internal-external structural coordination," constructs an indicator system for high-quality development in the insurance industry. The aim is to explore the evolution of high-quality development in the insurance industry.

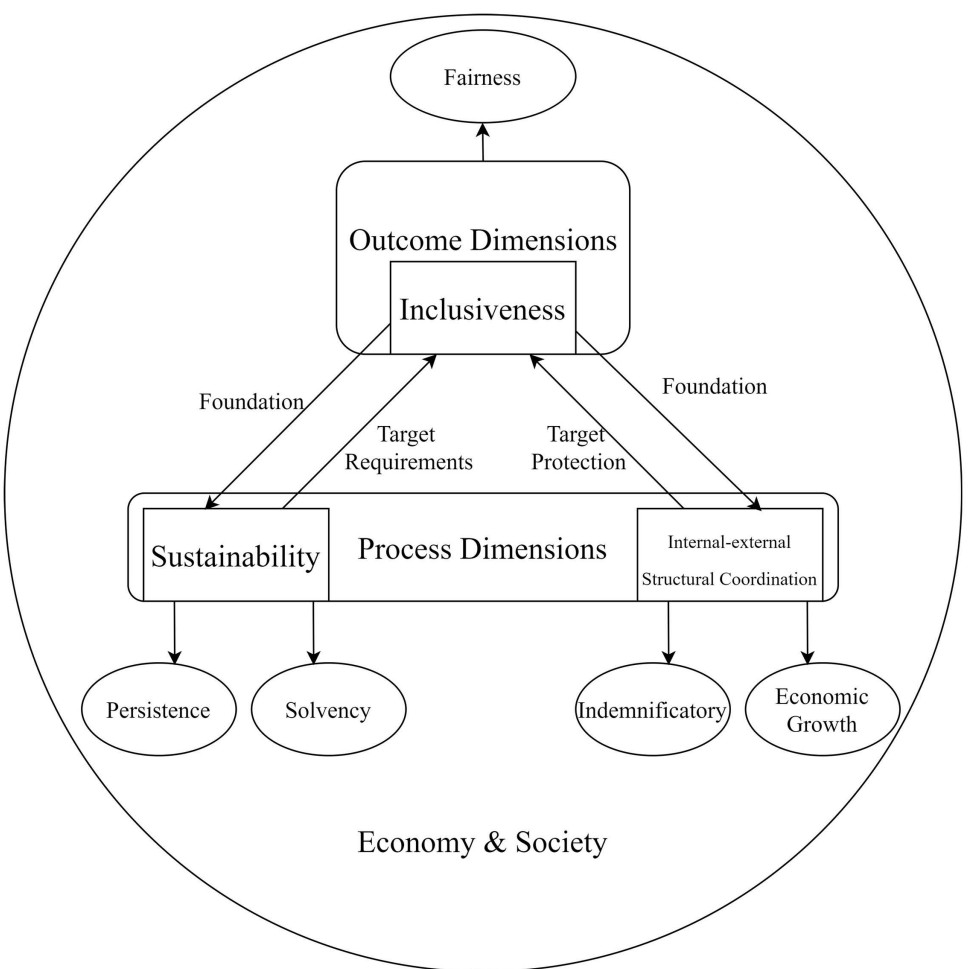

**Fig 1. Theoretical framework for high-quality development of the insurance industry.**

(1) The "Inclusiveness" dimension draws on the measurement experience of regional inclusive insurance and rural financial levels by Li et al. [5] and Zhang et al. [40]. The measurement is conducted at three levels: scale (number of insurance companies, number of insurance industry practitioners, premium income, life insurance premium income, property insurance premium income, ratio of insurance companies to local area, premium income per capita), coverage (insurance amount per local population, insurance amount per number of industrial enterprises, insurance density, insurance depth), and utility (property insurance claims per number of industrial enterprises, life insurance claims per local population, number of enterprises with insurance institutions holding shares).

(2) The "Sustainability" dimension is measured from the perspectives of business quality (province-level surrender amount per life insurance premium income) and operational quality (drawing inspiration from Gao Yiming's [35] analysis of the risk level of the financial industry at the provincial level, using the absolute values of the deviation of insurance claims and premium income from long-term trends).

(3) The "Internal-External Structural Coordination" dimension draws on the experiences of scholars such as Xiong [41,42], Gao [35], and Jiang [43]. It is measured at two levels: internal business structure coordination (non-auto insurance premium income to property insurance premium income, (claims + injury medical + annuity) to life insurance claims, (ordinary life insurance + accidental injury insurance + health insurance) to personal insurance income) and external market structure coordination (number of insurance intermediary companies, number of insurance technology companies, HHI index for life insurance, and HHI index for property insurance).

## Materials and methods

The key to formulating comprehensive indicators lies in assigning weights to the sub-indicators of each dimension. On one hand, it is essential to ensure that the data reflects its inherent objectivity and information content. On the other hand, the current research on the spatial distribution of the high-quality development index employs the entropy weight TOPSIS method. This study draws on the research by [1,44,45] that measures the level of high-quality development in cities. We use a combination of subjective analysis and entropy weighted TOPSIS method to measure the high-quality development level of the insurance industry in each province. Firstly, use the entropy weight TOPSIS method to assign weights to the indicators of the three dimensions, and the calculation method is as follows:

(1) Dimensionless processing of data:

$$Z_{ij}^{'} = \begin{cases} \dfrac{Z_{ij} - min_i}{max_i - min_i}, Z_{ij} \text{ is a positive indicator} \\ \dfrac{max_i - Z_{ij}}{max_i - min_i}, Z_{ij} \text{ is a negative indicator} \end{cases} \tag{1}$$

$Z_{ij}^{'}$ is the dimensionless data, $Z_{ij}$ indicates what is the i-th indicator in the j-th year, $min_i$ and $max_i$ represent the minimum and maximum values of the i-th indicator respectively.

(2) Construct a dimensionless matrix $P_{ij}$:

$$P_{ij} = \dfrac{Z_{ij}^{'}}{\sum_{i=1}^{n} Z_{ij}^{'}} \tag{2}$$

(3) Calculate the entropy of the indicator $e_j$:

$$e_j = -\ln(n)^{-1} \times \sum_{n}^{i=1} \left[ P_{ij} \times \ln\left(P_{ij}\right) \right] \tag{3}$$

(4) Calculate information utility $a_j$:

$$a_j = 1 - e_j \tag{4}$$

(5) Calculate weight $W_j$:

$$W_{ij} = \frac{a_j}{\sum_{j=1}^{n} a_j} \tag{5}$$

Subsequently, employing the subjective weighting method, each of the three dimensions is assigned a weight of 1/3, thereby computing the weight of each indicator as follows (Table 1).

Finally, the TOPSIS method is employed to calculate the high-quality development index of the insurance industry within each province. The calculation methodology is delineated as follows:

(6) Build a weighted matrix:

$$Y = \left(y_{ij}\right)_{mn} = \left[W_j \times Z_{ij}^{'}\right]_{mn} \tag{6}$$

(7) Determine the positive ideal value $Y^+$ and the negative ideal value $Y^-$:

$$\begin{cases} Y_j^+ = \left(y_1^+, y_2^+, y_3^+, y_4^+, \ldots, y_m^+\right), y_j^+ = max\,y_{ij} \\ Y_j^- = \left(y_1^-, y_2^-, y_3^-, y_4^-, \ldots, y_m^-\right), y_j^- = min\,y_{ij} \end{cases} \tag{7}$$

(8) Calculate Euclidean distance $D_i^+$ and $D_i^-$:

$$\begin{cases} D_i^+ = \sqrt{\sum_{n}^{j=1}(y_{ij} - y_j^+)^2} \\ D_i^- = \sqrt{\sum_{n}^{j=1}(y_{ij} - y_j^-)^2} \end{cases} \tag{8}$$

(9) Calculate ideal transfer progress C:

$$C = \frac{D_i^-}{D_i^- + D_i^+} \tag{9}$$

The C value spans from 0 to 1, where a larger C value signifies a heightened level of high-quality development within the insurance industry of the province.

## Data sources

The main data sources include the "Statistical Yearbooks" of various provinces in China from 2010 to 2020, the "Insurance Statistical Yearbook," and the websites of the financial regulatory

**Table 1. Index System for High Quality Development of the Insurance Industry.**

| First-level indexes | Second-level indexes | Third-level indexes (measure index) | weight | NO. |
|---|---|---|---|---|
| Inclusiveness | scale | Number of insurance companies (provincial-level and above) | 0.0259 | 1 |
| | | Number of insurance practitioners | 0.0252 | 2 |
| | | Premium income | 0.0250 | 3 |
| | | Life insurance premium income | 0.0249 | 4 |
| | | Property insurance premium income | 0.0253 | 5 |
| | | Number of Insurance company/local area | 0.0203 | 6 |
| | | Premium income/population | 0.0254 | 7 |
| | Coverage | Insurance amount/local population | 0.0185 | 8 |
| | | Insurance amount/number of local industrial enterprises | 0.0190 | 9 |
| | | Insurance density | 0.0255 | 10 |
| | | Insurance depth | 0.0263 | 11 |
| | Usability | Property insurance claims/number of industrial enterprises | 0.0229 | 12 |
| | | Life insurance claims/number of local population | 0.0251 | 13 |
| | | Number of enterprises with insurance institutions holding shares | 0.0241 | 14 |
| Sustainability | Business quality | Province-level surrender amount/ life insurance premium income | 0.1112 | 15 |
| | Operational quality | Absolute values of the deviation of insurance claims | 0.1111 | 16 |
| | | Absolute values of the deviation of premium income from long-term trends | 0.1110 | 17 |
| Internal-External Structural Coordination | Internal business structure coordination | Non-auto insurance premium income/property insurance premium income | 0.0484 | 18 |
| | | (Claims + injury medical + annuity)/ life insurance claims | 0.0483 | 19 |
| | | (Odinary life insurance + accidental injury insurance + health insurance)/ personal insurance income | 0.0490 | 20 |
| | External market structure coordination | Number of insurance intermediary companies, | 0.0453 | 21 |
| | | Number of insurance technology companies | 0.0427 | 22 |
| | | HHI index for life insurance | 0.0498 | 23 |
| | | HHI index for property insurance | 0.0499 | 24 |

bureaus of each province. Some data require processing, and the details are as follows: (1) Due to differences in the statistical scope of insurance institutions among provinces in terms of time and space, rendering them incomparable, the number of insurance companies at the provincial level and above is selected as the indicator. The data on the number of insurance companies in each province are sourced from the provincial "Financial Operation Reports" and the regional version of the "Insurance Statistical Yearbook."(2) Data on the number of insurance industry employees are sourced from the Chinese Labor Economic Database in the EPS database, specifically the number of urban insurance industry employees at the end of the year. (3) For some provinces and years with missing insurance amount data, linear interpolation after logarithmic transformation of insurance amounts is conducted to complete the data. (4) Due to the lack of data on the number of legal entities in each province before 2013, the number of industrial enterprises in each province is selected as a proxy indicator based on data availability and comparability. The data are sourced from the Wind database. (5) Data on the number of enterprises held by insurance institutions in each province are sourced from

the China National Research Data Sharing Platform (CNRDS), specifically from databases on shareholding of listed companies and basic information of listed companies. (6) Long-term trends are calculated using the HP filter method. Since the selected premium income, claims, and benefits paid are annual data, a parameter $\lambda=100$ is chosen based on previous scholars' experiences to calculate the long-term trend. The deviation between the data for the current year and the long-term trend is considered as the deviation data. (7) The number of insurance technology companies is obtained by searching relevant keywords such as "insurance technology" and "internet insurance" (8) The HHI (Herfindahl-Hirschman Index) is calculated based on the market share of premium income of each insurance company in each province, as provided in the "Insurance Statistical Yearbook."

## Results

Next, this paper will first conduct descriptive statistical analysis on the indices of the three major dimensions. During this process, the 31 provinces (excluding Hong Kong, Macau, Taiwan due to data unavailability) will be divided into three regions: eastern, central, and western China. This approach will provide an overall view of the high-quality development level of the insurance industry in China.

Following this, various methods will be employed to analyze the spatial distribution differences in the high-quality development of China's insurance industry from 2010 to 2015 across four aspects: Temporal Distribution, Spatial Distribution, Dynamic Transition, and Spatial Disparities. The methods include non-parametric kernel density estimation, the non-standard deviation ellipse method, traditional and spatial Markov chains and the Moran's Index.

### Descriptive statistics

Utilizing the aforementioned methodology, we selected and calculated the high-quality development index of the insurance industry for China's 31 provincial-level administrative regions (see Table 2). These provincial-level administrative regions are categorized into three major regions: east, central, and west. The eastern region comprises Beijing, Tianjin, Hebei, Liaoning, and Shanghai, along with 11 additional provincial-level administrative regions including Jiangsu, Zhejiang, Fujian, Shandong, Guangdong, and Hainan. The central region encompasses Shanxi, Jilin, Heilongjiang, Anhui, Jiangxi, Henan, Hubei, and Hunan. The western region includes Inner Mongolia Autonomous Region, Guangxi, Chongqing, Sichuan, and Guizhou, as well as 12 other provincial-level administrative regions: Yunnan, Shaanxi, Gansu, Qinghai, Ningxia, Xinjiang, and Xizang. Descriptive statistics detailing the results are depicted in Fig 2.

The high-quality development of China's insurance industry exhibits three characteristics:(1) China's high-quality development level in the insurance industry has shown a long-term upward trend, with only a decline observed during the three-year period from 2010 to 2012, and the most significant decline occurring in the central region.(2) There is a significant regional disparity in the high-quality development level of China's insurance industry. The

**Table 2. Provincial and regional divisions.**

| Regions | East | central | west |
|---|---|---|---|
| Provincial-level administrative regions | Beijing, Tianjin, Hebei, Liaoning, Shanghai, Jiangsu, Zhejiang, Fujian, Shandong, Guangdong, Hainan | Shanxi, Jilin, Heilongjiang, Anhui, Jiangxi, Henan, Hubei, Hunan | Inner Mongolia Autonomous Region, Guangxi, Chongqing, Sichuan, Guizhou, Yunnan, Shaanxi, Gansu, Qinghai, Ningxia, Xinjiang, Xizang |

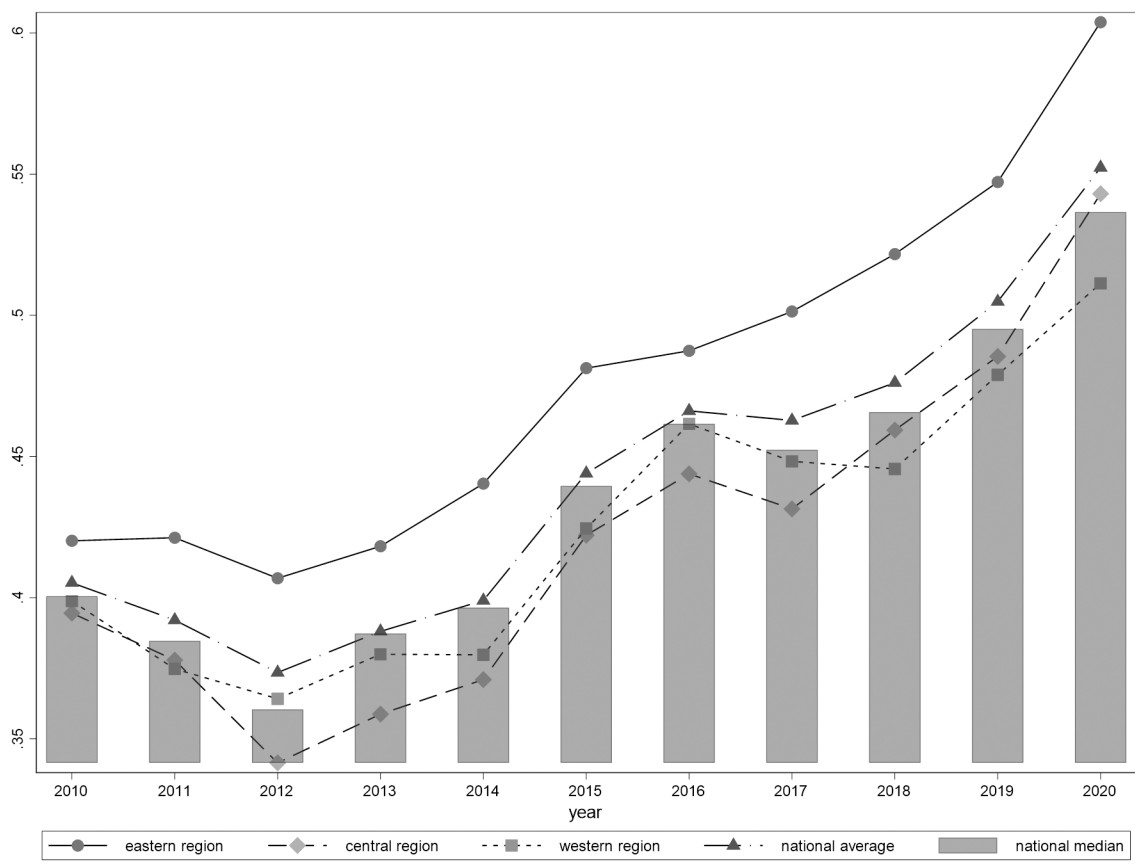

**Fig 2. Index of high-quality development of insurance industry.**

national average has consistently been higher than the national median over the years. It can be inferred that regions with higher development levels play a significant "pull-up" role. If we only observe from the average level, it may mask the true situation of regions with lower levels of high-quality development in the insurance industry, failing to reflect the real high-quality development level. (3) The regional differences in the high-quality development level of China's insurance industry are mainly reflected in the eastern region compared to the central and western regions. On the one hand, the high-quality development level of the insurance industry in the central and western regions is lower than the national average and national median levels. On the other hand, the eastern region's high-quality development level in the insurance industry is significantly higher than that of the central and western regions. However, the difference in the high-quality development level of the insurance industry between the central and western regions is not substantial and exhibits cross-regional characteristics (Fig 2).

Furthermore, the comprehensive index divided into three dimensions: inclusiveness, sustainability, and internal-external structural coordination. This allows for a more in-depth analysis of the reasons behind the changes in the high-quality development of the insurance industry (Fig 3).

The inclusiveness dimension index shows an overall upward trend. However, during the period from 2010 to 2012, there was a slow increase, and in the central and western regions, there was even a downward trend. This is closely related to the fact that in 2011, China's insurance depth and density were lower than the corresponding GDP growth level [46]. In general,

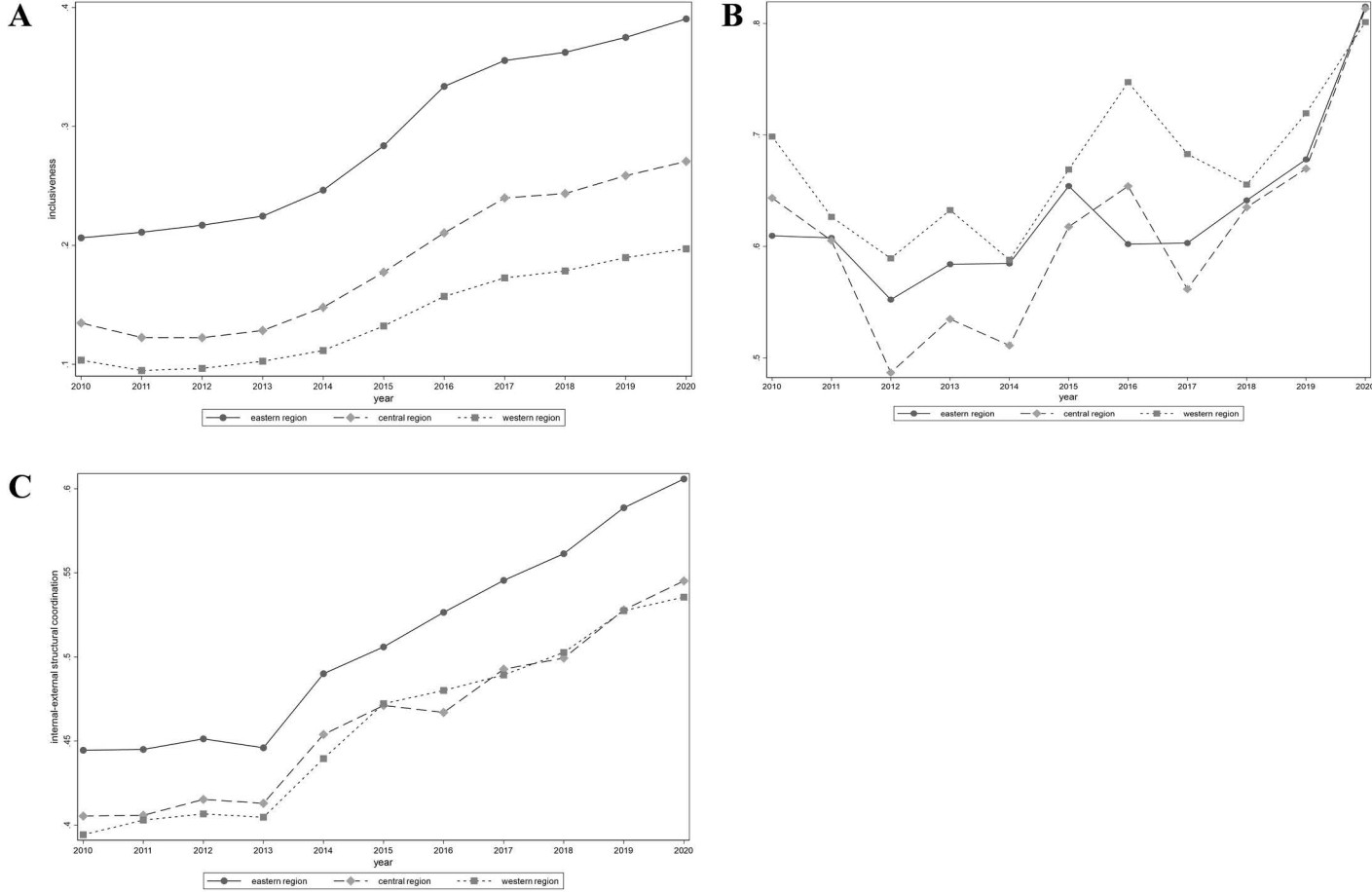

**Fig 3. Index of high-quality development of insurance industry by dimensions.** (A) respectively illustrate the inclusiveness dimension of high-quality development of insurance industry, (B) illustrate the sustainability dimension of high-quality development of insurance industry, (C) illustrate the internal-external structural coordination dimension of high-quality development of insurance industry.

it can be divided into three stages: a slow growth period from 2010 to 2012. On the one hand, this was influenced by internal factors such as bottlenecks in sales models and a lack of competition in mainland insurance products. On the other hand, external factors such as tight monetary policy, high running rates of interest and CPI, and the impact of the China Banking Regulatory Commission's new policy on bank agency channels [47]. The period from 2013 to 2016 was a period of rapid development. During this time, the life insurance rate marketization reform took place. After the removal of the "2.5% upper limit restriction" on life insurance in 2013, the China Insurance Regulatory Commission successively issued more than ten policies for life insurance rate reform, leading to rapid development in the insurance industry. The period from 2017 to 2020 was a stage of regulated development. With the implementation of the "Compensation Generation" in 2016 and the release of the "Notice on Regulating the Product Development and Design Behavior of Life Insurance Companies" on May 7, 2017, strengthening the regulation of the insurance industry, the industry's scale growth slowed down, entering a stage of stable growth.

The sustainability dimension index shows more obvious characteristics of volatility and sensitivity. In general, the periods of change in the sustainability dimension index overlap with those of the inclusiveness dimension index. Firstly, from 2010 to 2012, the sustainability

dimension index experienced a decline influenced by internal and external shocks. Secondly, during the period from 2012 to 2017, there was a period of fluctuating growth. Although policies such as the implementation of the "Compensation Generation" in 2016 and the release of the "Notice on Regulating the Product Development and Design Behavior of Life Insurance Companies" in 2017 were introduced, the effects of these policies were lagging. Therefore, their impact began to show after 2018. Lastly, from 2018 to 2020, the sustainability dimension index began to steadily rise, and regional differences among the three major regions diminished, officially entering a stage of stable development.

The internal-external structural coordination dimension is similar to the comprehensive index. On the one hand, in terms of trend characteristics, it shows a long-term upward trend, but around 2013 is a turning point. In addition to being influenced by internal and external shocks, this turning point may also be attributed to the establishment of the first domestic internet insurance company in 2013, marking the beginning of rapid development in China's insurance technology [48]. On the other hand, the index in the eastern region is much higher than that in the central and western regions. However, the difference between the central and western regions is not significant. The reason is that the central region has a larger proportion of bonus insurance in life insurance and a larger proportion of motor vehicle insurance in property insurance. Additionally, the western region has a larger proportion of agriculture in the three major industrial structures, and agricultural insurance has developed well in the insurance products in the western region [24].

In summary, the trend features of the comprehensive index mainly exhibit characteristics of the inclusiveness and internal-external structural coordination dimensions. The fluctuation characteristics of a certain stage are more prominently reflected in the sustainability dimension. The high-quality development index of the insurance industry can effectively demonstrate the high-quality development status of the insurance industry at a certain stage and the impact of influencing factors. Therefore, constructing this index has practical significance.

### Temporal distribution

The kernel density estimation method is a non-parametric approach used to estimate the probability density of random variables. It is characterized by its robustness and minimal reliance on specific model assumptions. Consequently, this paper employs the KDE method to estimate the probability density distribution of random variables, which is particularly useful for analyzing the insurance industry. This method allows for an examination of the time distribution, variability, and distribution shape of mass development. The following represents the estimated form of the kernel density function:

$$f\left(rf\right) = \frac{1}{nh}\sum_{n}^{j=1} K\left(\frac{rf_j - \overline{rf}}{h}\right) \tag{10}$$

$$K\left(rf\right) = \frac{1}{\sqrt{2\pi}} e^{-\frac{rf_j^2}{2}} \tag{11}$$

$rf$ is the high−quality development index of the insurance industry in each province. $\overline{rf}$ is the average value for high-quality development of the insurance industry. $n$ is the number of observations. $h$ is the bandwidth. The selection of bandwidth is based on the trade−off between the smoothness and accuracy of the density function. This article adopts the "rule of thumb" to determine. $rf_j$ represents independent and identically distributed observations. $K(.)$ is the selection of kernel density function. The kernel density functions selected for

kernel density estimation generally include Gaussian kernel function, gamma kernel function and uniform kernel function. This paper chooses Gaussian kernel function for estimation.

The three-dimensional kernel density plot of the high-quality development in the national insurance industry. Firstly, examining the dynamic evolution path of the "main peak," it shows a trajectory of "leftward-rightward" movement. This indicates that during this period, the overall level of high-quality development in the national insurance industry experienced a brief "decline" followed by an "upturn," suggesting a temporary downturn during 2010-2012 due to comprehensive national impacts. Moreover, in terms of distribution, it exhibits a characteristic of rightward extension, signifying that the high-quality development levels of most provinces nationwide are relatively low. Provinces with high levels show significant differences, while those with lower levels exhibit smaller differences, resulting in a substantial overall disparity (Fig 4).

Furthermore, during the period of 2010-2020, there were multiple occurrences of "double peaks" or even "triple peaks," indicating a "multimodal" phenomenon. This suggests a gradient effect and differentiation effect in the high-quality development levels of the insurance industry. Provinces with higher development levels, represented by Beijing, Guangdong, Shanghai, etc., cluster around a comprehensive index of 0.65, while those in the central and western regions cluster around a comprehensive index of 0.5. This demonstrates a spatial gradient feature in the high-quality development levels of the national insurance industry, with high-level regions represented by Guangdong and Shanghai and medium to low-level regions represented by the central and western regions. This phenomenon may be attributed to significant economic disparities between regions. As an industry, the development level of insurance is influenced by the overall economic conditions of a region. The economic development gap between the eastern and the central and western regions has led to differences in the levels of development within the insurance industry across these areas.

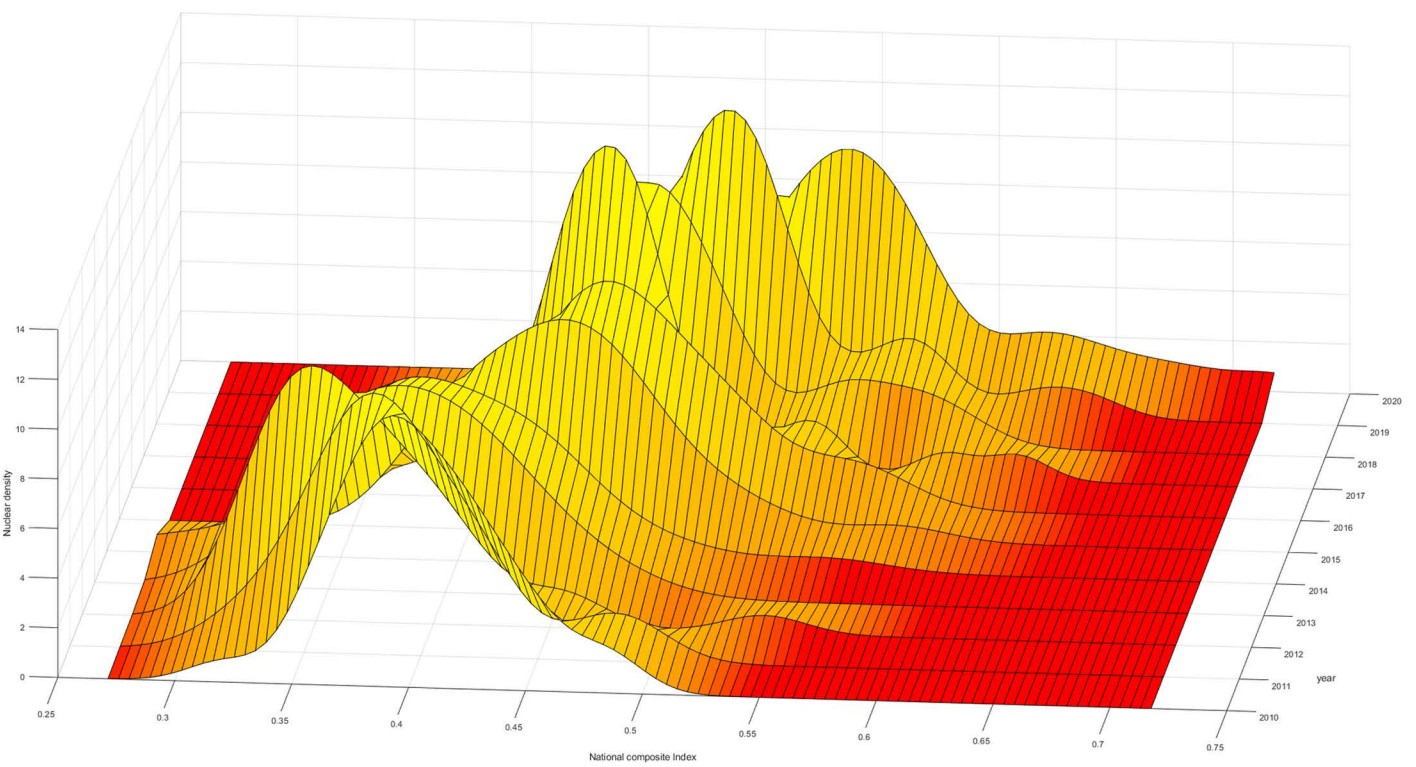

**Fig 4. National Comprehensive Index Kernel Density Map.**

To further analyze the changes in the high-quality development levels of the insurance industry within the three major regions, Firstly, looking at the aggregation interval and movement trajectory of the "main peak," the comprehensive index aggregation interval in the eastern region is larger than that in the central and western regions. Therefore, the development level in the eastern region is higher than that in the central and western regions, and all three regions experience a "decline-upturn" phase. Furthermore, in terms of "kurtosis," the eastern region evolves towards "no peak," indicating that the high-quality development levels of various provinces within the eastern region gradually diverge, with internal development level differences far greater than those in the central and western regions. Even though the high-quality development level in the eastern region is much higher than that in the central and western regions, it is limited to certain provinces such as Guangdong and Shanghai, while provinces like Hainan and Hebei need improvement in the high-quality development of the insurance industry. This phenomenon may be related to differences in the economic structures of various regions. Provinces like Guangdong and Shanghai, which serve as economic centers, host numerous companies and thus experience a high demand for insurance services. In contrast, provinces such as Hainan, where tourism is the primary industry, have relatively lower insurance demand due to the different economic focus. In contrast, the central and western regions evolve towards "single peak aggregation," indicating that the internal differences in the high-quality development levels of the insurance industry in the central and western regions are gradually narrowing. Lastly, in terms of distribution, all three regions exhibit a feature of "leftward extension," indicating an evolution of high-quality development levels within the regions towards higher local levels (Figs 5–7).

The observed phenomenon can be attributed to several factors: (1) Disparities in Economic Development Levels: Significant differences exist in the economic development levels among

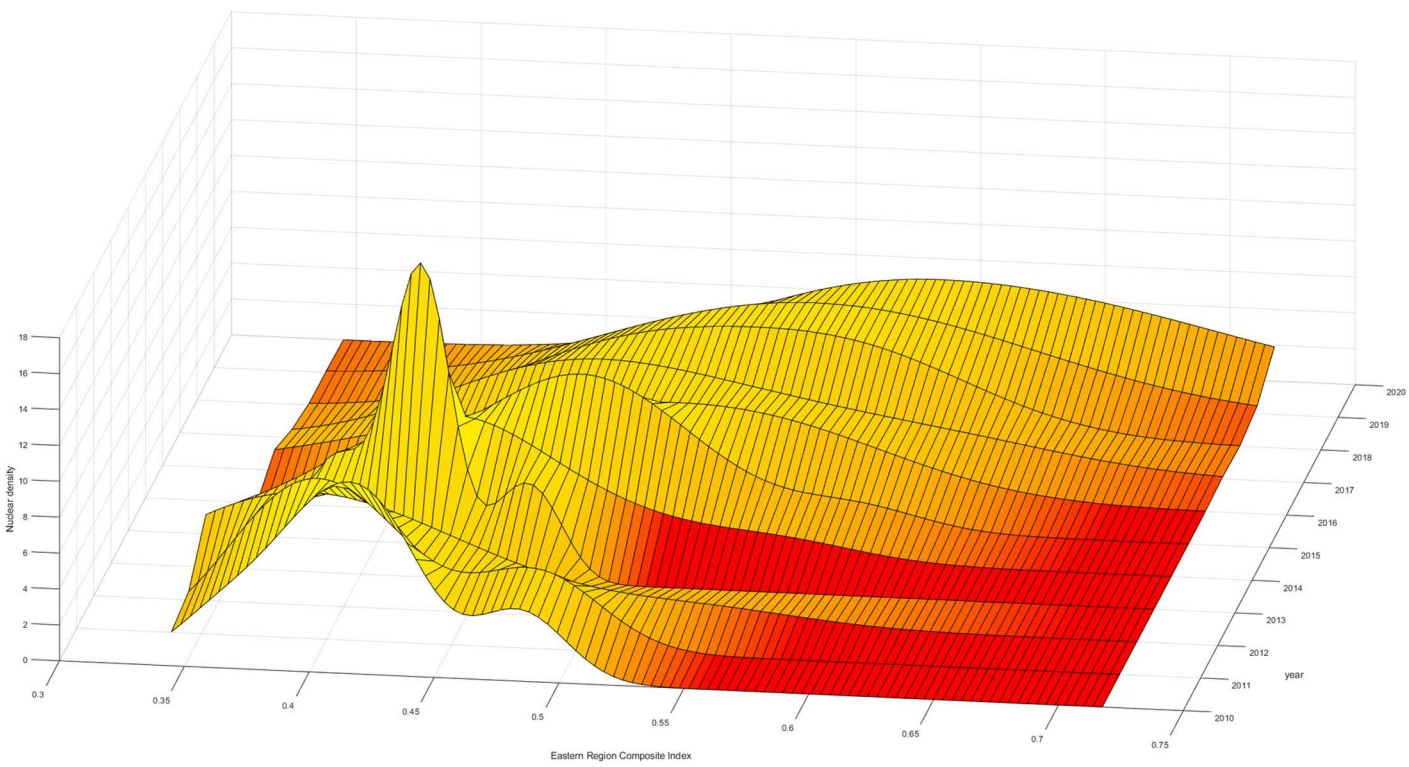

**Fig 5. Eastern Comprehensive Index Kernel Density Map.**

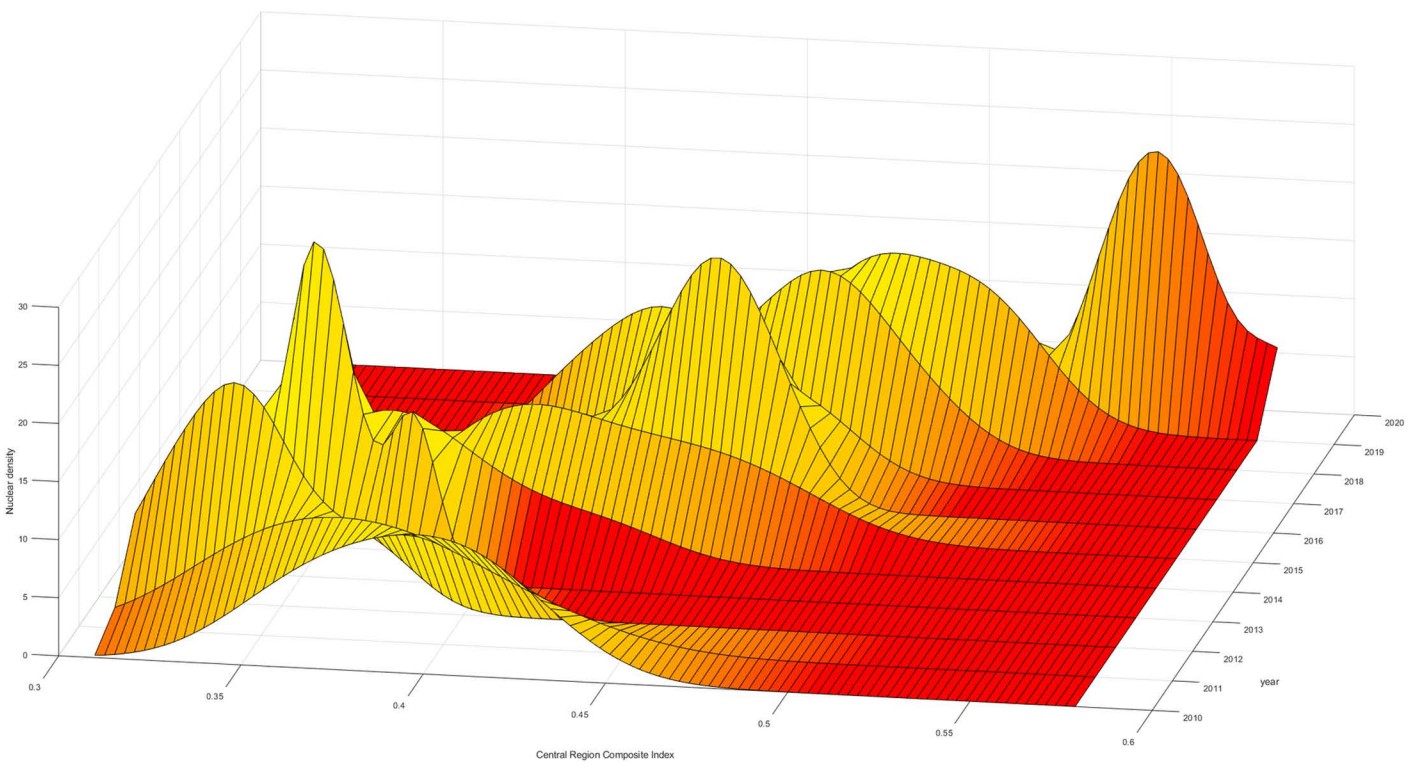

**Fig 6. Central Comprehensive Index Kernel Density Map.**

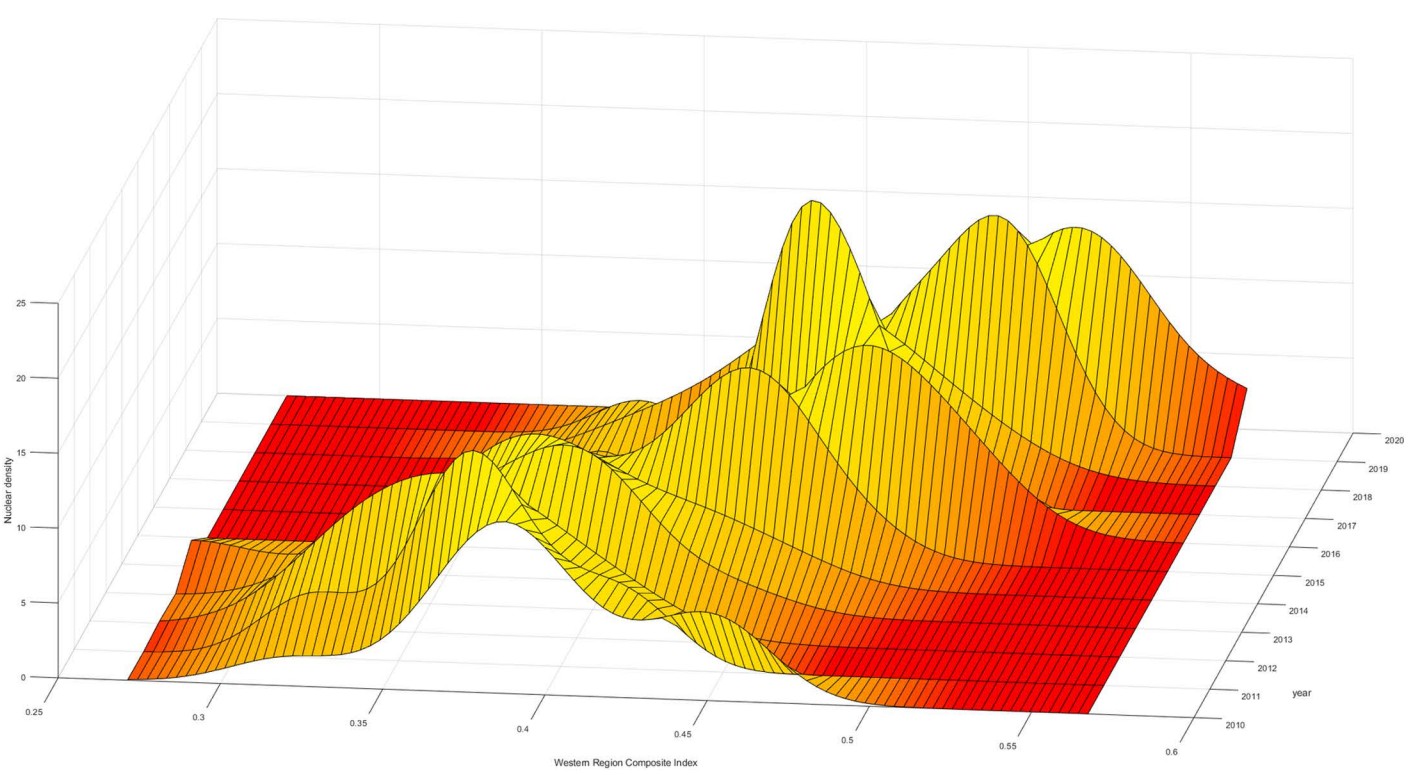

**Fig 7. West Comprehensive Index Kernel Density Map.**

the eastern, central, and western regions. According to financial development theory [49], the demand for financial products, including insurance, tends to follow economic growth. With the eastern region experiencing faster economic growth compared to the central and western regions in recent years, the demand for insurance products has correspondingly increased. (2) Impact of Urban Agglomeration: Urban agglomeration has accentuated the gradient effect across the eastern, central, and western regions. Progressive reforms to the household registration system from 2010 to 2020 facilitated population mobility, thereby accelerating urban agglomeration. The majority of urban regional economies are concentrated in the eastern region, whereas the central and western regions still predominantly feature rural economies. This acceleration of agglomeration in the eastern region has resulted in a noticeable gradient phenomenon.(3) Variations in Economic Structures: Economic structural differences within regions also contribute to disparities in insurance industry development. For instance, while both Hainan and Guangdong belong to the eastern region, Hainan's economic development relies heavily on tourism, which may not directly support substantial growth in the insurance industry. Conversely, Guangdong boasts unique industrial clusters that provide robust support for regional insurance industry development, serving as a significant economic backbone. In conclusion, the national high-quality development levels of the insurance industry show positive trends. The eastern region needs to address the significant internal differences in development levels, and while the central and western regions are witnessing a reduction in differences, addressing the overall development level differences will be a challenge in the next stage.

## Spatial distribution

In this study, we chose to employ the Standard Deviation Ellipse (SDE) method to investigate the overall dynamic changes in the spatial distribution characteristics of high-quality development in the national insurance industry. The distinctive feature of the SDE method lies in its ability to visually depict economic elements in space through the parameters of center, major axis, minor axis, and orientation angle. It provides an intuitive representation of the development direction, trends, and changes in the relative distribution range of the research object in the overall economic space during a specific period. This approach quantitatively describes the dynamic evolution trends of the overall spatial distribution of economic elements, including characteristics such as centrality, dispersion, directionality, and spatial morphology. Therefore, we can more intuitively illustrate the distribution changes in the relative level of high-quality development of China's insurance industry across the country. The calculation formula used for this purpose is as follows:

$$\bar{X} = \frac{\sum_{i=1}^{n} w_i x_i}{\sum_{i=1}^{n} w_i} \qquad (12)$$

$$\bar{Y} = \frac{\sum_{i=1}^{n} w_i y_i}{\sum_{i=1}^{n} w_i} \qquad (13)$$

$$\tan\theta = \frac{\left(\sum_{i=1}^{n} w_i^2 \tilde{x}_i^2 - \sum_{i=1}^{n} w_i^2 \tilde{y}_i^2\right) + \sqrt{(\sum_{i=1}^{n} w_i^2 \tilde{x}_i^2 - \sum_{i=1}^{n} w_i^2 \tilde{y}_i^2)^2 + 4\sum_{i=1}^{n} w_i^2 \tilde{x}_i^2 \tilde{y}_i^2}}{2\sum_{i=1}^{n} w_i^2 \tilde{x}_i \tilde{y}_i} \qquad (14)$$

$$\sigma_x = \sqrt{\dfrac{\sum_{i=1}^{n}\left(w_i \tilde{x}_i \cos\theta - w_i \tilde{y}_i \sin\theta\right)^2}{\sum_{i=1}^{n} w_i^2}} \tag{15}$$

$$\sigma_y = \sqrt{\dfrac{\sum_{i=1}^{n}\left(w_i \tilde{x}_i \sin\theta - w_i \tilde{y}_i \cos\theta\right)^2}{\sum_{i=1}^{n} w_i^2}} \tag{16}$$

$x_i$ and $y_i$ are the abscissa and ordinate of the area respectively, $\overline{X}$ and $\overline{Y}$ are respectively the abscissa and ordinate of the ellipse center. θ represents the azimuth angle of the ellipse, that is, the angle from which the long axis deviates from the north axis. $\sigma_x$ and $\sigma_y$ represents the length along the x and y axes respectively.

The software used for this analysis is ArcGIS 10.8. Based on the Standard Deviation Ellipse maps of the comprehensive index and three major dimension indices for the years 2010 to 2020. As shown in "X-axis" and "Y-axis" of Table 3, the overall spatial distribution exhibits an "East (slightly North) – West (slightly South)" pattern. It indicates that the driving force for the spatial distribution of the insurance industry in China is greater in the "East–West" direction, with a larger difference between the "East" and "West" than between the "South" and "North." Based on "Center of gravity longitude"、"Center of gravity latitude"、"X-axis"、"Y-axis" and "Perimeter" of Table 3 (For specific map display, please ask the author).The regions covered by the Standard Deviation Ellipse generally include the central and southern parts of Inner Mongolia, areas north of Guangdong Province, east of Sichuan Province, and regions west of the Yellow Sea. This implies that the main areas of high-quality development in the insurance industry are located in the eastern regions (excluding the three northeastern provinces), some central regions, and a few western regions. Indeed, the observed phenomenon aligns with the regional disparities in economic policies implemented since China's reform and opening up. Following the initiation of "reform and opening up", the eastern coastal areas have garnered more significant policy support, particularly in terms of urban development initiatives such as the establishment of special economic zones, the Yangtze River Delta urban agglomeration, and the Beijing-Tianjin-Hebei urban agglomeration. These policies have contributed to accentuating the divergence between the eastern and western regions, surpassing the disparities between the northern and southern regions.

**Table 3. Basic parameters of standard deviation ellipse.**

| Year | Weight dimension | Perimeter | Center of gravity longitude | Center of gravity latitude | X-axis length | Y-axis length |
|------|-----------------|-----------|----------------------------|----------------------------|---------------|---------------|
| **2010** | Comprehensive | 74.0948 | 111.7235 | 33.5869 | 13.8544 | 9.5309 |
| **2015** | Comprehensive | 74.1625 | 111.7380 | 33.4796 | 13.8437 | 9.5677 |
| **2020** | Comprehensive | 73.0087 | 112.0510 | 33.5285 | 13.6755 | 9.3620 |
| **2010** | Inclusiveness | 66.0580 | 113.8214 | 33.9299 | 12.1015 | 8.7947 |
| **2015** | Inclusiveness | 67.6984 | 113.7690 | 33.7043 | 12.3268 | 9.1011 |
| **2020** | Inclusiveness | 66.9562 | 113.7228 | 33.8322 | 12.2621 | 8.9190 |
| **2010** | Sustainability | 75.1501 | 111.2011 | 33.5309 | 14.0321 | 9.6903 |
| **2015** | Sustainability | 75.2773 | 111.1699 | 33.3903 | 14.0471 | 9.7172 |
| **2020** | Sustainability | 74.1979 | 111.6017 | 33.4683 | 13.9739 | 9.4226 |
| **2010** | Structural | 74.6154 | 111.7971 | 33.5526 | 14.0203 | 9.5148 |
| **2015** | Structural | 74.6176 | 111.6723 | 33.5079 | 14.0272 | 9.5073 |
| **2020** | Structural | 73.7463 | 111.8516 | 33.4614 | 13.8040 | 9.4682 |

## Dynamic transition

This study employs traditional Markov chains and spatial Markov chains to evaluate the dynamic transition patterns of high-quality development levels in the national insurance industry. The Markov chain can be utilized to discretize continuous attribute values of regional phenomena across different periods. By employing data classification methods, these values are divided into $k$ levels, allowing for the measurement of probability distributions and changes across these levels. The evolutionary development phenomenon is approximated as a Markov process. The calculation formula is as follows:

$$M_{ij} = \frac{n_{ij}}{n_i} \tag{17}$$

$M_{ij}$ represents the probability value of a province with high-quality development level $i$ in the insurance industry at time $t$ converted to level $j$ $n_{ij}$ represents the total number of provinces at level $i$ at time $t$ that are converted to level $j$ at time $t+1$ $n_i$ represents the total number of $i$ level provinces at all times during the study period. Using a quartile division, the high-quality development index of each province is categorized into four levels. All indices from 2010 to 2020 are divided into four groups according to quartiles (see Table 4). The first group is with the index lower than 0.3871, the second group is with the index between 0.3871 and 0.4363, and the third group is between 0.4363 and 0.4829. The fourth group has an index higher than 0.4829. However, traditional Markov chains do not account for the interdependence and correlation between regional economies. To address this limitation, the spatial Markov chain incorporates the concept of "spatial lag". The spatial Markov chain examines the neighbor region status of the spatial unit by adding a spatial weight matrix. $M_{ij}(k)$ represents the probability of transitioning to level $k$ represents the probability of transitioning to level $j$ at

**Table 4. Calculation results of traditional Markov transition matrix and spatial Markov transition matrix.**

| Spatial lag type | t/t+1 | I | II | III | IV | Frequency |
|---|---|---|---|---|---|---|
| **No-lag** | I | 0.6353 | 0.3294 | 0.0353 | 0 | 85 |
| | II | 0.1977 | 0.3488 | 0.3837 | 0.0698 | 86 |
| | III | 0.0357 | 0.1548 | 0.4524 | 0.3571 | 84 |
| | IV | 0 | 0 | 0.1273 | 0.8727 | 55 |
| I | I | 0.6809 | 0.3191 | 0 | 0 | 47 |
| | II | 0.2609 | 0.3478 | 0.3478 | 0.0435 | 23 |
| | III | 0.2 | 0.4 | 0.2 | 0.2 | 5 |
| | IV | 0 | 0 | 0.2 | 0.8 | 5 |
| II | I | 0.6129 | 0.3548 | 0.0323 | 0 | 31 |
| | II | 0.2857 | 0.3714 | 0.2286 | 0.1143 | 35 |
| | III | 0.0476 | 0.1905 | 0.5714 | 0.1905 | 21 |
| | IV | 0 | 0 | 0 | 1 | 3 |
| III | I | 0.6 | 0.2 | 0.2 | 0 | 5 |
| | II | 0.0435 | 0.3478 | 0.6087 | 0 | 23 |
| | III | 0 | 0.1351 | 0.5135 | 0.3514 | 37 |
| | IV | 0 | 0 | 0.2 | 0.8 | 25 |
| IV | I | 0 | 0.5 | 0.5 | 0 | 2 |
| | II | 0 | 0.2 | 0.6 | 0.2 | 5 |
| | III | 0.0476 | 0.0952 | 0.2857 | 0.5714 | 21 |
| | IV | 0 | 0 | 0.0455 | 0.9545 | 22 |

time $t+1$ under the condition that the spatial lag type is $k$. The spatial lag type is determined by the spatial lag value. The spatial lag value is the spatially weighted average of the attribute values of the spatial neighbor areas. The calculation formula is:

$$Lag = Y_i W_{ij} \tag{18}$$

$Y_i$ is the attribute value of the spatial unit. $W_{ij}$ is the j-th element in the i-th row of the spatial weight matrix $W$. The transition matrices for both traditional Markov and spatial Markov (with adjacency matrix as the spatial weight matrix) are computed using Matlab, and the results are presented below.

The first type, without spatial lag, is the traditional Markov transition matrix. Firstly, the probabilities on the diagonal maintain the original level, namely 0.6353, 0.3488, 0.4524, and 0.8727. Except for the diagonal elements in the second row, the probabilities on the diagonal are greater than those off the diagonal, indicating a significant probability of maintaining the original levels I, III, and IV, demonstrating a "self-locking" effect. However, the "self-locking" effect of level II is not strong, with a probability of transitioning from level II to level III being 0.3837, which is higher than the diagonal elements. This suggests a likelihood of promoting the development of the insurance industry in the next stage. Furthermore, analyzing the probabilities adjacent to the diagonal, indicating the probability of a province's insurance industry high-quality development level rising or falling by one level in the next stage, it is observed that, except for level IV, the probability of each province's insurance industry rising by one level is greater than the probability of falling by one level. However, the probability of rising from level I to level II is smaller than the probability of other levels rising by one level, implying that provinces with lower levels of high-quality development in the insurance industry face greater difficulties initially. This phenomenon may be attributed to the fact that in the early stages of insurance industry development, there is often a lack of mature infrastructure, economic foundation, and policy support. These factors can create significant challenges and hinder the industry's growth. Still, after rising one level, the probability of advancing to level II is higher. Once a solid economic foundation is established, it fosters a positive interaction between insurance development and economic growth. As a result, the subsequent stages of development tend to encounter fewer difficulties compared to the initial stage, due to the supportive infrastructure and economic stability. Therefore, provinces with lower levels in the insurance industry need to seize the opportunity to improve their high-quality development and quickly enter the "comfort zone" of advancement. Finally, for provinces at level IV in the high-quality development of the insurance industry, there is a probability of "regression" in the next stage, with a probability of 0.1272 regressing to level III. Hence, provinces with a higher level of high-quality development in the insurance industry need to focus on maintaining their current development achievements to prevent regression in the high-quality development level.

However, the traditional Markov transition matrix does not consider the spatial correlation of economic and social factors in geographic space, particularly whether there is a correlation between the high-quality development levels of the insurance industry among adjacent provinces. By observing the spatial lag types I, II, III, and IV in the spatial transition matrix, significant differences are found compared to the traditional Markov transition matrix. Therefore, provinces with different high-quality development levels in the insurance industry have an impact on the high-quality development level of the province. Firstly, provinces at level II in high-quality development show a significant probability of advancing to the next level when adjacent to provinces at levels I, II, III, and IV, with probabilities of 0.3478, 0.2286, 0.6087, and 0.6, respectively. This indicates a substantial likelihood of advancing one level, and provinces

at level II need to seize the opportunity to improve the high-quality development level of their insurance industry. Provinces at level I, although having a strong "self-locking" effect, also need to take advantage of opportunities to advance to level II, creating more opportunities for progress in high-quality development of the insurance industry.

Certainly, the observed phenomenon can be attributed to various factors. In the nascent stages of the insurance industry's development, challenges such as limited income among residents and insufficient economic support may have hindered its growth. However, as the industry matured and attained a certain level of development, increased income levels and economic prosperity facilitated greater adoption of insurance products. Furthermore, there exists a symbiotic relationship between the development of the insurance industry and economic growth. As the insurance sector expands, it not only provides financial protection to individuals and businesses but also fosters economic stability. This, in turn, stimulates further economic growth and income generation, thereby creating a positive feedback loop. As a result, this stage of development experiences rapid growth propelled by mutually reinforcing dynamics between the insurance industry, economic growth, and income levels. Furthermore, provinces at different levels of high-quality development show a clear "driving" effect when adjacent. Provinces at levels III and IV have a more pronounced "driving" effect on neighboring provinces' high-quality development levels, with provinces at level II adjacent to provinces at level III having the probability of high-quality development level advancing from 0.2286 to 0.6087. Similarly, provinces at level III adjacent to provinces at level IV have the probability of high-quality development level advancing from 0.3514 to 0.5714, showing a significant "driving" effect. When provinces at levels I and II are adjacent, the probability of high-quality development level advancing increases from 0.3191 to 0.3548. It may be that fostering collaboration and resource sharing among neighboring regions can significantly drive the development of the insurance industry. Leveraging each other's advantages in terms of income, talent, equipment, technology, and other resources can create synergies that benefit all parties involved. Lastly, under the spatial Markov chain, provinces at level IV in high-quality development may still experience a "regression" in the insurance industry. Therefore, maintaining the stability of the high-quality development levels among regions is of great significance. Based on the descriptive statistics presented in the Descriptive Statistics section, it is plausible that this phenomenon may be attributed to fluctuations in the "sustainability" dimension during certain periods. In some instances, the insurance industry may have experienced rapid growth by employing strategies such as reducing prices, lowering entry barriers, or increasing interest rates, which could lead to significant instability. Prior to strengthened regulatory measures in China in 2016, substantial fluctuations were observed, which might have contributed to the "regression" of the insurance industry in certain regions.

## Spatial disparities

This article analyzes the high-quality development levels of the national insurance industry from two perspectives: spatial correlation and disparity. Firstly, given the geographical, economic, and social interactions among neighboring provinces, it is essential to explore whether these interactions affect the independence of high-quality development in the insurance industry across provinces. Secondly, as there are regional economic development disparities, it is necessary to analyze the differences in the development of the insurance industry across provinces. To address these aspects, this study employs Moran's Index to elucidate the correlation and disparity in the high-quality development of the insurance industry across provinces.

**Moran's index.** This study opts to utilize provincial data, as the correlation among provinces is weaker than that among cities. The choice of an adjacency matrix as a spatial weight matrix is made to measure the Moran's Index for the three dimensions of the comprehensive index for the high-quality development of the insurance industry. The spatial autocorrelation of the insurance

industry indicates that provinces with higher levels of insurance industry development tend to be clustered together, as do provinces with lower levels. This clustering reflects a pattern where regions with similar levels of insurance industry development are geographically proximate. The calculation formula for spatial autocorrelation is as follows:

$$I = \frac{\sum_{i=1}^{n} \sum_{j=1}^{n} w_{ij} \left( x_i - \bar{x} \right) \left( x_j - \bar{x} \right)}{S^2 \sum_{i=1}^{n} \sum_{j=1}^{n} w_{ij}} \tag{19}$$

$$S^2 = \frac{\sum_{i=1}^{n} \left( x_i - \bar{x} \right)^2}{n} \tag{20}$$

Among them, $I$ is the Moran index, which ranges from -1 to 1. The closer to the absolute value of 1, the greater the spatial autocorrelation. $S^2$ represents the sample variance, and $w_{ij}$ represents the spatial weight matrix. The results are presented below:

From the calculation of Moran's Index across three major dimensions (see Tables 5–7), it is evident that these dimensions exhibit distinct characteristics in spatial correlation. The dimensions of "inclusiveness" and "internal-external structural coordination" exhibit significant spatial autocorrelation.

**Table 5. Calculation Results of Moran Index for Inclusiveness.**

| Variables | I | E(I) | sd(I) | z | p-value* |
|---|---|---|---|---|---|
| 2010 | 0.136 | -0.033 | 0.092 | 1.840 | 0.066 |
| 2011 | 0.137 | -0.033 | 0.094 | 1.811 | 0.070 |
| 2012 | 0.161 | -0.033 | 0.095 | 2.045 | 0.041 |
| 2013 | 0.176 | -0.033 | 0.095 | 2.195 | 0.028 |
| 2014 | 0.210 | -0.033 | 0.095 | 2.565 | 0.010 |
| 2015 | 0.247 | -0.033 | 0.095 | 2.964 | 0.003 |
| 2016 | 0.238 | -0.033 | 0.095 | 2.863 | 0.004 |
| 2017 | 0.243 | -0.033 | 0.095 | 2.915 | 0.004 |
| 2018 | 0.222 | -0.033 | 0.095 | 2.681 | 0.007 |
| 2019 | 0.219 | -0.033 | 0.096 | 2.642 | 0.008 |
| 2020 | 0.223 | -0.033 | 0.096 | 2.680 | 0.007 |

**Table 6. Calculation Results of Moran Index for Sustainability.**

| Variables | I | E(I) | sd(I) | z | p-value* |
|---|---|---|---|---|---|
| 2010 | 0.041 | -0.033 | 0.096 | 0.773 | 0.440 |
| 2011 | -0.141 | -0.033 | 0.096 | -1.127 | 0.260 |
| 2012 | 0.052 | -0.033 | 0.097 | 0.878 | 0.380 |
| 2013 | 0.204 | -0.033 | 0.097 | 2.441 | 0.015 |
| 2014 | 0.129 | -0.033 | 0.095 | 1.708 | 0.088 |
| 2015 | 0.002 | -0.033 | 0.098 | 0.363 | 0.717 |
| 2016 | 0.288 | -0.033 | 0.096 | 3.336 | 0.001 |
| 2017 | 0.144 | -0.033 | 0.098 | 1.808 | 0.071 |
| 2018 | 0.006 | -0.033 | 0.096 | 0.408 | 0.683 |
| 2019 | 0.063 | -0.033 | 0.097 | 0.996 | 0.319 |
| 2020 | 0.001 | -0.033 | 0.096 | 0.355 | 0.722 |

**Table 7. Calculation Results of Moran Index for Internal-External Structural Coordination.**

| Variables | I | E(I) | sd(I) | z | p-value[*] |
|-----------|------|--------|--------|-------|-----------|
| **2010** | 0.142 | -0.033 | 0.095 | 1.849 | 0.064 |
| **2011** | 0.105 | -0.033 | 0.092 | 1.502 | 0.133 |
| **2012** | 0.126 | -0.033 | 0.092 | 1.732 | 0.083 |
| **2013** | 0.101 | -0.033 | 0.089 | 1.516 | 0.130 |
| **2014** | 0.178 | -0.033 | 0.091 | 2.334 | 0.020 |
| **2015** | 0.080 | -0.033 | 0.090 | 1.252 | 0.211 |
| **2016** | 0.183 | -0.033 | 0.092 | 2.342 | 0.019 |
| **2017** | 0.185 | -0.033 | 0.091 | 2.411 | 0.016 |
| **2018** | 0.147 | -0.033 | 0.090 | 1.999 | 0.046 |
| **2019** | 0.136 | -0.033 | 0.091 | 1.860 | 0.063 |
| **2020** | 0.159 | -0.033 | 0.093 | 2.077 | 0.038 |

From the perspective of "inclusiveness," the observed pattern may be due to the interactive nature of adjacent provinces' economies. The development of the insurance industry in surrounding areas can enhance economic growth in neighboring provinces, which in turn stimulates local economic development and promotes the growth of the local insurance industry. The concept of "internal-external structural coordination" may arise from the similarity in economic structures between adjacent regions. The development structure of the insurance industry often aligns with the regional economic structure, leading to similarities in the insurance industry structure across neighboring areas. Specifically, during the period from 2010 to 2020, the Moran's Index for the "inclusiveness" dimension ranged between 0.136 and 0.243, with a majority of the years being significant at the 5% level, indicating the strongest spatial autocorrelation. Similarly, the Moran's Index for the "internal-external structural coordination" dimension ranged from 0.08 to 0.185, with significance observed in all years except for 2011, 2013, and 2015 at the 10% level.

**Local moran's index.** To ensure robustness, we choose the Local Moran's Index and use 2020 data for robustness recommendations. Consistent with the findings in the Moran's Index section, the majority of regions for the year 2020 exhibit distribution in the first and third quadrants for the dimensions of "inclusiveness" and "internal-external structural coordination," indicating significant positive spatial correlation between regions (Figs 8–10).

From the above analysis, it can be inferred that there exists a correlation between the dimensions of "inclusiveness" and "internal-external structural coordination" in the high-quality development of the insurance industry across regions. This correlation may be attributed to the following reasons: Firstly, neighboring regions share similarities in resident preferences, income levels, and informational elements, which provide advantages for the expansion and development of the insurance industry in surrounding areas. Secondly, there are similarities in the economic structures of neighboring regions, leading to a correlation between the structure of the insurance industry (especially insurance product structure) and the economic structure. For instance, regions in the central and western parts of the country have a larger distribution of agriculture, resulting in a greater demand for agricultural insurance and thereby creating structural similarities in the insurance industry.

## Discussion

In the above part, we have systematically analyzed the spatial and temporal distribution status and dynamic evolution of high quality in China's insurance industry. In the next part, we will further use The Dagum Gini coefficient to analyze the spatial differences, differential changes

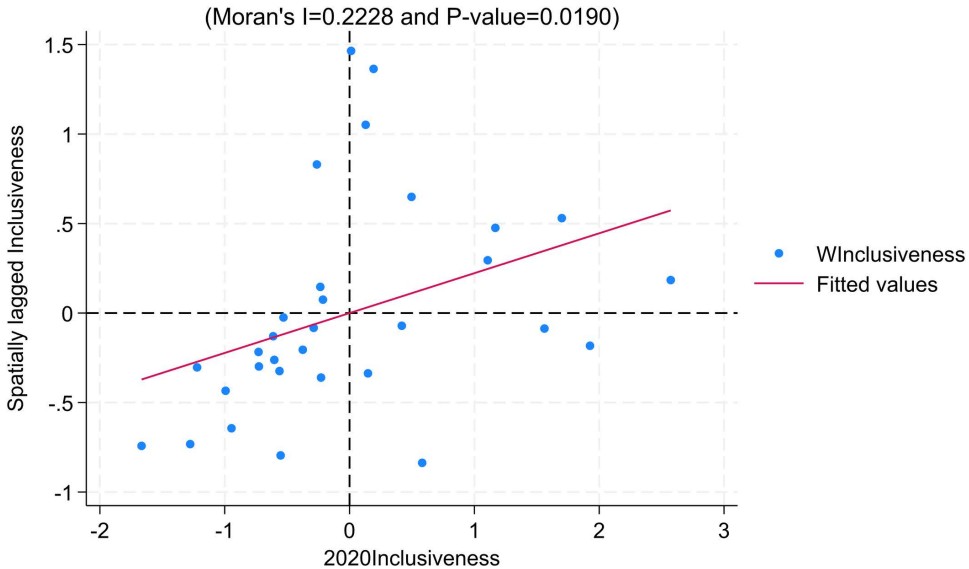

**Fig 8. Moran Scatter Plot of Inclusiveness.**

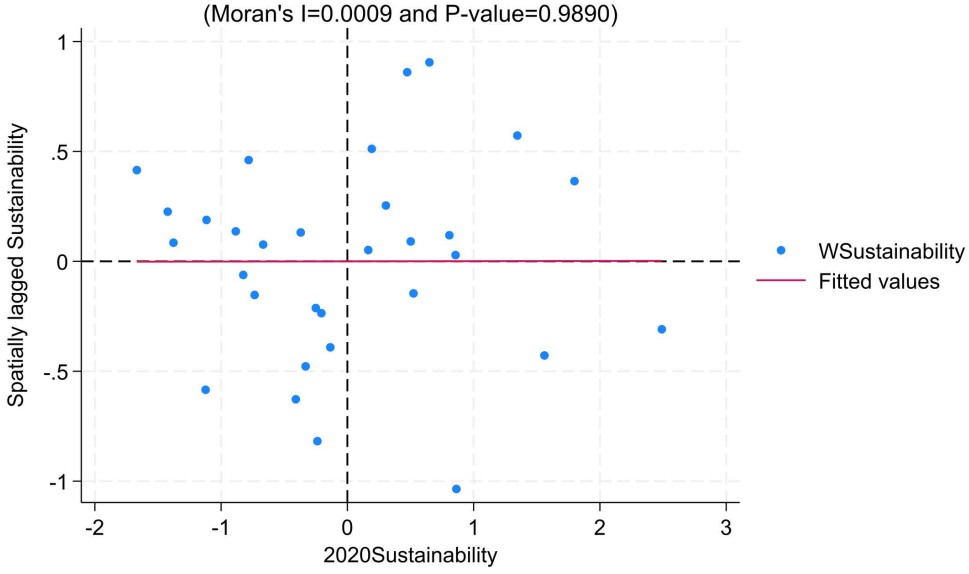

**Fig 9. Moran Scatter Plot of Sustainability.**

and main differentiating factors in the distribution of China's insurance industry. The Dagum Gini coefficient decomposition method is an improved approach that allows the calculation of the Gini coefficient according to sub-samples. In this study, the Dagum Gini coefficient is calculated using the composite index of high-quality development levels in the insurance industry at the provincial level as sub-samples. This method enables the analysis of spatial disparities in the high-quality development levels of the insurance industry at both the provincial and regional levels. Furthermore, through the analysis of contributions to the super-variation density, it is possible to understand the cross-overlapping situation of high-quality

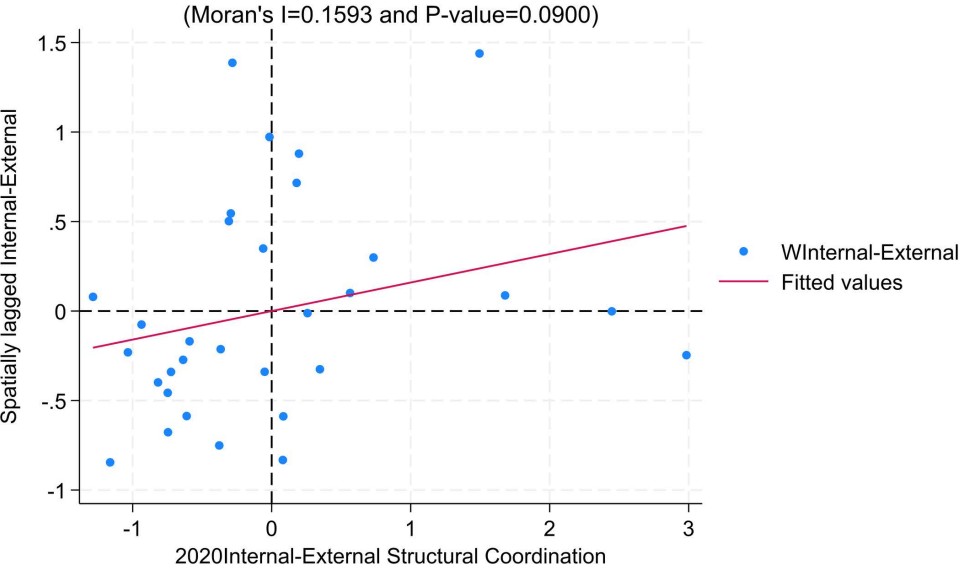

**Fig 10. Moran Scatter Plot of Internal-External Structural Coordination.**

development in the insurance industry in the eastern, central, and western regions. The results of the Dagum Gini coefficient decomposition for the years 2010, 2013, 2015, 2018, and 2020 are presented below:

Based on Table 8, the overall Gini coefficient shows an upward trend from 2010 to 2020, indicating a gradual widening of the disparity in the high-quality development levels of the insurance industry among provinces. Although there is a slight downward trend in 2020 compared to 2018, the change is not significant. Therefore, coordinated development of the insurance industry at the provincial level in China still requires effort.

Decomposing the overall Gini coefficient into intra-regional differences, inter-regional differences, and contributions from super-variation density, it is observed that inter-regional differences have been consistently high, showing an increasing trend. The gap in the high-quality development of the insurance industry among the eastern, central, and western regions is

**Table 8. Calculation Results of Dagum Gini coefficient decomposition.**

| Dagum Gini coefficient | | 2010 | 2013 | 2015 | 2018 | 2020 |
|---|---|---|---|---|---|---|
| **Overall Gini coefficient** | | 0.0496 | 0.0545 | 0.0559 | 0.0569 | 0.0551 |
| **Decomposition items and contributions** | **Intra-Regional differences** | 0.0162 | 0.0140 | 0.0151 | 0.0144 | 0.0131 |
| | Contribution | 32.6279% | 25.6721% | 27.0683% | 25.2409% | 23.7617% |
| | Regional differences | 0.0141 | 0.0330 | 0.0303 | 0.0368 | 0.0389 |
| | Contribution | 28.3658% | 60.6038% | 54.2484% | 64.6727% | 70.4943% |
| | Supervariable density | 0.0194 | 0.0075 | 0.0104 | 0.0057 | 0.0032 |
| | Contribution | 39.0063% | 13.7242% | 18.6833% | 10.0864% | 5.7440% |
| **Intra-Regional differences** | Eastern | 0.0492 | 0.0417 | 0.0528 | 0.0516 | 0.0552 |
| | Central | 0.0519 | 0.0471 | 0.0387 | 0.0267 | 0.0211 |
| | Western | 0.0327 | 0.0231 | 0.0376 | 0.0394 | 0.0295 |
| **Regional differences** | Eastern | 0.0562 | 0.0612 | 0.0698 | 0.0678 | 0.0598 |
| | Central | 0.0488 | 0.0786 | 0.0725 | 0.0822 | 0.0846 |
| | Western | 0.0452 | 0.0463 | 0.0403 | 0.0360 | 0.0375 |

widening, with inter-regional differences being the main factor causing disparities in the high-quality development levels of the insurance industry in China. Considering Fig 2, this is mainly due to the significantly higher level of high-quality development in the eastern region compared to the central and western regions. Additionally, intra-regional difference levels show a gradual decline but with insignificant changes. The contribution from super-variation density displays a substantial decrease in variation; the number of provinces with overlapping high-quality development levels is decreasing within each region. Even provinces with lower high-quality development levels in the eastern region are gradually surpassing provinces with higher levels in the central and western regions. This underscores the increasing regionalization of the high-quality development of the insurance industry nationwide, further confirming the "gradient effect" in the high-quality development of the insurance industry, where the regional level can be roughly indicative of the province's high-quality development level, hindering balanced development of the insurance industry between regions.

The characteristics of intra-regional differences in the high-quality development of the insurance industry are markedly different. The Gini coefficient in the eastern region is not only at a high level but also shows an upward trend, consistent with the results presented in the kernel density map of high-quality development in the insurance industry in the eastern region in Fig 5. This indicates a widening disparity in the high-quality development of the insurance industry among provinces in the eastern region, even showing a "peak-less" feature after 2016. Some provinces, such as Beijing and Guangdong, represent a significantly higher level of high-quality development compared to other areas in the eastern region. However, when considering the contribution from super-variation density, even provinces with significant differences in the high-quality development of the insurance industry within the eastern region have levels surpassing provinces in the central and western regions. This implies a pronounced stepwise phenomenon in the high-quality development of the insurance industry in China: the Gini coefficient of high-quality development in the insurance industry in the eastern region is rapidly and steadily decreasing; the central region exhibits a gradually narrowing disparity, indicating a coordinated feature; the western region's Gini coefficient remains roughly stable between 0.03 and 0.04, with low and stable variation.

Regarding inter-regional differences, disparities between the eastern and central regions have remained consistently high, showing an "expansion—contraction" pattern, albeit gradually narrowing after 2015. However, it still remains at a relatively high level. Disparities between the eastern and western regions have shown a rapid increase over the 11-year period, with a slight decrease in 2015. The Gini coefficient of inter-regional differences in 2018 exceeded 0.8, far higher than the difference between the eastern and central regions. Differences between the central and western regions show a fluctuating downward trend with relatively small variations, consistent with the results shown in Fig 2. The differences between the eastern and central regions are substantial, while the differences between the central and western regions are small.

## Conclusions

This paper elucidates the theoretical connotations of high-quality development in the insurance industry through three dimensions: "Inclusiveness," "Sustainability," and "Internal-External Structural Coherence." Based on these dimensions, we construct an evaluation index system for the high-quality development of the insurance industry. The spatial and temporal distribution characteristics of China's insurance industry development are analyzed, and the conclusions are as follows:

(1) Regarding the "Inclusiveness" dimension, the overall level of high-quality development in China's insurance industry shows an upward trend. Particularly, since the initiation of

market-oriented reforms in life insurance premium rates in 2013, the "Inclusiveness" index has been consistently rising. It is noteworthy that during the period from 2010 to 2020, the development speed of the "Inclusiveness" dimension in the insurance industry was observed as "East> Central> West." The Western region exhibits relatively low levels of "Inclusiveness" development compared to the East and Central regions, indicating significant regional disparities persist.

(2) From the perspective of the "Sustainability" dimension, the early years witnessed an overall unstable characteristic in the insurance industry due to imperfect institutional frameworks in China's insurance market and regulation. This suggests a serious issue of extensive operations during this period. It wasn't until the implementation of the "Compensation Second Generation" in 2016, coupled with regulatory strengthening measures like the May 7, 2017, announcement by the China Insurance Regulatory Commission on regulating the product development behavior of life insurance companies, that the Chinese insurance market gradually became regulated, and regional fluctuations diminished.

(3) Examining the "Internal-External Structural Coherence" dimension, overall stability has gradually improved, particularly after 2013. Influenced by factors such as "Broadband China," insurance technology companies, and the market-oriented reforms in life insurance premium rates, the "Internal-External Structural Coherence" dimension of the insurance industry entered an accelerated development phase. However, there remains a significant disparity between the development levels in the Eastern region compared to the Central and Western regions. Therefore, in the next phase, the Central and Western regions need to pay attention to coordinating the development of product entities and market entities.

(4) From an overall perspective, the issue of development disparities between regions in China is more pronounced than disparities within regions, highlighting conspicuous inequalities in the regional development of the insurance industry. While the Eastern region exhibits a relatively high level of insurance industry development, substantial inter-provincial variations create evident "inequities" within the region. On the other hand, although internal disparities in the development levels of the insurance industry are gradually diminishing in the Central and Western regions, the overall development level remains relatively low.

Therefore, the following recommendations are given in regard to the conclusions:

(1) Grasp commonalities and individual differences: Recognize the market position and functional roles of the insurance industry in socialism with Chinese characteristics. Tailor strategies for high-quality development based on the unique challenges faced by different regions. Insurance companies in the eastern region prioritize innovation in insurance products, such as "insurance + medical services," to cater to the diverse insurance needs of the population. In contrast, companies in the central and western regions focus more on promoting insurance products, particularly targeting low-income groups, to increase accessibility and coverage. On the other hand, regions with lower levels should focus on being the "stabilizer" of economic development, as economic development plays a decisive role in the high-quality development of the insurance industry. This involves constructing a highly adaptable, competitive, and inclusive modern insurance system, leveraging insurance technology for digital services, and enhancing the insurance industry's ability to serve private enterprises, thereby promoting high-quality development.

(2) Understand the interconnectedness of high-quality development among regions: Leverage the development of the digital economy and the level of digitization in the insurance industry to facilitate the flow of information between regions. Rational use of information diffusion can improve the awareness of insurance and the service levels of insurance institutions

in economically lagging regions. Deepen cooperation across regions, build an integrated insurance market, achieve interconnectivity of infrastructure, remove administrative barriers, and overcome institutional obstacles to drive high-quality development in less developed regions. Leveraging the urban agglomeration construction policy as an opportunity, we will collaborate with economic entities in surrounding areas to expand insurance demand. At the same time, we will fully utilize digital infrastructure to drive the digital transformation of the insurance industry. Establish specialized insurance institutions that cater to regional-specific insurance demands. This involves constructing specialized insurance institutions with regional advantages to drive the development of related products.

(3) Improve the construction of the index system for the high-quality development of the insurance industry: A comprehensive system for the high-quality development of the insurance industry provides a solid foundation and guidance for real development, offering direction for the high-quality development of the insurance industry at the regional level. Due to the theoretical and practical characteristics of "high-quality development," the requirements for the high-quality development of the insurance industry will continue to evolve with changes in the economy and society. It is crucial to constantly monitor changes in economic and social demands for insurance to break away from the past development concept of "speed only" and combine speed with quality, and long-term with short-term. This encourages and supports the insurance industry to return to its essence.

Our research focuses on examining the high-quality development of China's insurance industry from a macro perspective, aiming to form a comprehensive understanding of its current state. In the next stage, we will adopt a micro perspective, studying the high-quality development of China's insurance industry with a focus on individual insurance companies. This approach will further enrich and complement our research findings.

## Supporting information

**S1 file. All supplementary materials are available in the file titled "Supporting information.zip".1. Data Files**: The file **"data.zip"** contains all the data used in this article.**(1) File "all_data.dta"**: Data for all basic indicators. **(2) File "edge_matrix.dta"**: Spatial weight matrix data required for spatial analysis. **(3)Files "High-Quality.dta"**, **"Inclusiveness.dta"**, **"Sustainability.dta"**, and **"Internal-External.dta"**: Represent data for high-quality development, the "inclusiveness dimension," the "sustainability dimension," and the "internal-external structural coordination" dimension, respectively. **2. Code Files**: The file **"code.zip"** contains all the model codes required for operations in this article. **(1) File "code.do"**: Code for the entropy weight TOPSIS model and Moran's index. **(2) File "insurance_ker.m"**: Code for the non-parametric kernel density model. **(3) Files "tra_mark.m"** and **"space_mark.m"**: Custom functions codes for traditional Markov chains and spatial Markov chains, respectively. **(4) File "mar.m"**: Running code for Markov chains. **(5) File "Dagum_Gini.m"**: Custom functions code for calculating the Dagum Gini coefficient. **(6) File "Dagum_compute.m"**: Code for running the calculation of the Dagum Gini coefficient.
(ZIP)

## Author contributions

**Conceptualization:** Heng Zhu, Meimei Tang.

**Investigation:** Bo Xiong, Meimei Tang.

**Writing – original draft:** Heng Zhu, Bo Xiong.

**Writing – review & editing:** Heng Zhu, Meimei Tang.

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
