## [Decision Letter · Decision Letter 0]

22 May 2024

PONE-D-24-01860A Study on the Evaluation and Spatiotemporal Distribution Characteristics of the High-Quality Development Level of China's Insurance IndustryPLOS ONE

Dear Dr. ZHU,

Thank you for submitting your manuscript to PLOS ONE. After careful consideration, we feel that it has merit but does not fully meet PLOS ONE’s publication criteria as it currently stands. Therefore, we invite you to submit a revised version of the manuscript that addresses the points raised during the review process.

**This study evaluates China's insurance sector development based on inclusiveness, sustainability, and structural coordination. It highlights significant regional disparities and imbalance, with the eastern region showing higher development but internal inequities, while the central and western regions demonstrate gradual convergence but subdued overall benchmarks. We commend the authors for their comprehensive study on evaluating China's insurance sector development. However, several key points need addressing:**

**1.In the last paragraph of the introduction, correct "Daugm's" to "Dagum's" Gini coefficient.**

**2.Introduce a new section titled "Materials and Methods" (Part 3), emphasizing the construction of the indicator system, along with data sources and calculation methodologies.**

**3.Merge the second half of Part 3 and Part 4 into the RESULTS section for better organization, focusing on the analysis and spatial distribution of high-quality development.**

**4.Clarify the methodology for dividing provinces into four levels based on quartile division and ensure consistency in table references (e.g., middle of 2013 and 2018 as 2015).**

**5.Consider extending the analysis in Section 4.4.2 with local Moran's index for a more comprehensive examination.**

**6.Incorporate a discussion section to provide insights and interpretations of the results.**

**7.Verify the format of figures depicting China's map and correct any discrepancies. Additionally, ensure accuracy in references and include international paper references.**

**Overall, a major revision is recommended to address these points thoroughly and enhance the clarity and coherence of the paper.**

**8.Table 7: Calculation Results of Dagum Gini coefficient. One of the year data in the Dagum Gini coefficient composition is incomplete. Please confirm**

**9.other details**

**(1).Introduction:**

Further refine the marginal contribution of the article. There have been some studies on the measurement of the level of high-quality development of the insurance industry in the literature. How does this article differentiate from existing literature? The article also mentions " Currently, there is a lack of research on the construction of a comprehensive evaluation system for high-quality development in the insurance industry." Where does this point reflect? It is recommended that the author further elaborate and explain.

(2).Literature Review:

The content of the literature review is not focused enough, especially in the first part. It is not necessary to discuss the economic foundation of high-quality development from scratch, but rather to focus more on the review and evaluation of research on the high-quality development of the insurance industry in recent years. This will highlight the position of this article in existing literature and its innovations.

(3).Indicator Construction:

The article defines the high-quality development of the insurance industry based on the dimensions of "development" and "quality" as " the industry's development over a certain period, leading to changes in the satisfaction level of consumers' needs and values for insurance products and services." The focus is on consumers, but the indicator system does not reflect the concept of "consumer satisfaction."

(4).The article should clearly explain the data sources for indicator construction.

We look forward to receiving your revised manuscript.

Kind regards,

Yu kun Wang

Academic Editor

PLOS ONE

Journal Requirements:

3. We note that Figures 8, 9, 10 and 11 in your submission contain map images which may be copyrighted. All PLOS content is published under the Creative Commons Attribution License (CC BY 4.0), which means that the manuscript, images, and Supporting Information files will be freely available online, and any third party is permitted to access, download, copy, distribute, and use these materials in any way, even commercially, with proper attribution. For these reasons, we cannot publish previously copyrighted maps or satellite images created using proprietary data, such as Google software (Google Maps, Street View, and Earth). For more information, see our copyright guidelines: http://journals.plos.org/plosone/s/licenses-and-copyright.

We require you to either present written permission from the copyright holder to publish these figures specifically under the CC BY 4.0 license, or remove the figures from your submission:

a. You may seek permission from the original copyright holder of Figures 8, 9, 10 and 11 to publish the content specifically under the CC BY 4.0 license.  

Reviewers' comments:

Reviewer's Responses to Questions

**Comments to the Author**

1. Is the manuscript technically sound, and do the data support the conclusions?

Reviewer #1: Yes

Reviewer #2: Partly

2. Has the statistical analysis been performed appropriately and rigorously? 

Reviewer #1: No

Reviewer #2: Yes

3. Have the authors made all data underlying the findings in their manuscript fully available?

Reviewer #1: Yes

Reviewer #2: Yes

4. Is the manuscript presented in an intelligible fashion and written in standard English?

Reviewer #1: No

Reviewer #2: Yes

5. Review Comments to the Author

**Reviewer #1: ** This study evaluates China's insurance sector development based on inclusiveness, sustainability, and structural coordination. It highlights significant regional disparities and imbalance, with the eastern region showing higher development but internal inequities, while the central and western regions demonstrate gradual convergence but subdued overall benchmarks. I commend the authors for their comprehensive study on evaluating China's insurance sector development. However, several key points need addressing:

1.In the last paragraph of the introduction, correct "Daugm's" to "Dagum's" Gini coefficient.

2.Introduce a new section titled "Materials and Methods" (Part 3), emphasizing the construction of the indicator system, along with data sources and calculation methodologies.

3.Merge the second half of Part 3 and Part 4 into the RESULTS section for better organization, focusing on the analysis and spatial distribution of high-quality development.

4.Clarify the methodology for dividing provinces into four levels based on quartile division and ensure consistency in table references (e.g., middle of 2013 and 2018 as 2015).

5.Consider extending the analysis in Section 4.4.2 with local Moran's index for a more comprehensive examination.

6.Incorporate a discussion section to provide insights and interpretations of the results.

7.Verify the format of figures depicting China's map and correct any discrepancies. Additionally, ensure accuracy in references and include international paper references.

Overall, a major revision is recommended to address these points thoroughly and enhance the clarity and coherence of the paper.

**Reviewer #2: ** This study, based on the criteria of "inclusiveness", "sustainability," and "internal-external structural coordination," establishes an evaluative framework for appraising the high-quality development of the insurance sector. It systematically gauges the overarching high-quality developmental status of China's insurance industry across regions. Employing non-parametric kernel density estimation, the Standard Deviation Ellipse, and spatial Markov chain, the investigation dynamically scrutinizes the national landscape of high-quality evolution within the insurance sector over the temporal spectrum. Furthermore, Moran's index and Dagum's Gini coefficient are harnessed to disentangle the spatial interdependence and heterogeneity characterizing the high-quality progression of the insurance industry among provinces. The findings disclose a pronounced regional development gap throughout China, surpassing intra-regional disparities and underscoring a notable concern of imbalance in regional insurance industry development.

How to evaluate high-quality development and its spatial evolution characteristics is an important and worthy topic of attention. The overall empirical analysis of the article is relatively standardized, and the writing is quite fluent. A considerable amount of work has been done in the construction of indicators and empirical analysis. However, there are also the following issues:

1.Introduction:

Further refine the marginal contribution of the article. There have been some studies on the measurement of the level of high-quality development of the insurance industry in the literature. How does this article differentiate from existing literature? The article also mentions " Currently, there is a lack of research on the construction of a comprehensive evaluation system for high-quality development in the insurance industry." Where does this point reflect? It is recommended that the author further elaborate and explain.

2.Literature Review:

The content of the literature review is not focused enough, especially in the first part. It is not necessary to discuss the economic foundation of high-quality development from scratch, but rather to focus more on the review and evaluation of research on the high-quality development of the insurance industry in recent years. This will highlight the position of this article in existing literature and its innovations.

3.Indicator Construction:

The article defines the high-quality development of the insurance industry based on the dimensions of "development" and "quality" as " the industry's development over a certain period, leading to changes in the satisfaction level of consumers' needs and values for insurance products and services." The focus is on consumers, but the indicator system does not reflect the concept of "consumer satisfaction."

4.Other Details:

The article should clearly explain the data sources for indicator construction.

6. PLOS authors have the option to publish the peer review history of their article (what does this mean? ). If published, this will include your full peer review and any attached files.

**Do you want your identity to be public for this peer review?** For information about this choice, including consent withdrawal, please see our Privacy Policy .

Reviewer #1: No

Reviewer #2: No

---

## [Author Response · Author response to Decision Letter 0]

10 Jun 2024

Dear Editor,

We sincerely thank you and all reviewers for the critical/valuable feedback. We feel lucky that our manuscript went to these reviewers as the valuable comments from them not only helped us with the improvement of our manuscript but suggested some neat ideas for future studies. Please do forward our heartfelt thanks to these experts.

Based on the comments we received, careful modifications have been made to the original manuscript. All changes were marked in yellow text and “Track Changes”. We hope the new manuscript will meet your standard. Below you will find out point-by-point responses to the comments/questions.

We look forward to hearing from you regarding our submission. We would be glad to respond to any further questions and comments that you may have.

Heng Zhu

(On behalf of all authors)

Response to comments on “Spatio-temporal Evolution and Distribution Characteristics of the High-Quality Development of China's Insurance Industry (PONE-D-24-01860)”

Response to Editor Comments

1.Please ensure that your manuscript meets PLOS ONE's style requirements, including those for file naming.

Thank you for your suggestion. We have checked the format of the manuscript and the file name.

2.Thank you for uploading your study's underlying data set. Unfortunately, the repository you have noted in your Data Availability statement does not qualify as an acceptable data repository according to PLOS's standards.

Thank you for your suggestion. We have organized all the data and code and uploaded them to the submission system.

3. We note that Figures 8, 9, 10 and 11 in your submission contain map images which may be copyrighted. All PLOS content is published under the Creative Commons Attribution License (CC BY 4.0), which means that the manuscript, images, and Supporting Information files will be freely available online, and any third party is permitted to access, download, copy, distribute, and use these materials in any way, even commercially, with proper attribution. For these reasons, we cannot publish previously copyrighted maps or satellite images created using proprietary data, such as Google software (Google Maps, Street View, and Earth).

Thank you for your suggestion. Due to the graphic format and copyright issues of Chinese maps, we have decided to delete map and submit only the "Basic Parameters of Standard Deviation Ellipse" table（Table 2 in the manuscript）. This table quantifies the results of the standard deviation ellipse method. While it is not as visually intuitive as directly displaying the map, the same conclusions can be drawn based on the specific data changes presented. If readers require the map results, we can provide them upon request.

Response to Reviewer 1 Comments

1.In the last paragraph of the introduction, correct "Daugm's" to "Dagum's" Gini coefficient.

Thanks for your careful checks. We acknowledge the oversight in our initial submission, where "Dagum's" was incorrectly written. We have corrected this error in the last paragraph of the introduction.

2.Introduce a new section titled "Materials and Methods" (Part 3), emphasizing the construction of the indicator system, along with data sources and calculation methodologies.

Thank you for your suggestion. In response to your request, we have added the "Materials and Methods" section to Part 3. This section details the calculation process for weighting each indicator and explains the data sources. Due to issues such as missing data in certain years and inconsistent statistical standards, some data required separate processing and calculations. Consequently, we have included detailed explanations and methods addressing the reasons for and processes used in handling such data. Meanwhile, we have attached the calculation code for comprehensive indicators.

3.Merge the second half of Part 3 and Part 4 into the RESULTS section for better organization, focusing on the analysis and spatial distribution of high-quality development.

Thank you for your suggestion. We have restructured Part 3, designating its second section as 4.1 to present the descriptive statistical results. Additionally, we have included an explanation of the method used to classify the 31 provinces into the East, Central, and West regions.（This classification is supported by the province division code, detailed in the "Dividing into regions" section of the Stata code. ）

4.Clarify the methodology for dividing provinces into four levels based on quartile division and ensure consistency in table references (e.g., middle of 2013 and 2018 as 2015).

Thank you for your suggestion. Our selection of quantiles was based on the overall data from 2010 to 2020, with quartiles selected for grouping (detailed quartiles have been added to line 493 of the article). We did not separately select quartiles from a representative year (such as 2014, 2015, or 2016) for grouping. The reasons are as follows:

Firstly, the interquartile range of the overall year is larger than that of a single year, as shown in Table 1, which can display the probability of changes in the high-quality development index of the insurance industry in each province on a larger scale. For example, Tables 2 and 3 show the traditional Markov matrix and spatial Markov matrix that divide the interquartile range based on 2015 and the overall year, respectively. The traditional Markov matrix includes regions that have developed from level I to level III. For the overall year, it has developed from below 0.3871 to 0.4363 to 0.4829, while for 2015, it has developed from 0.4140 to 0.4691. This indicates that the development level displayed is greater based on the overall year.

Secondly, the analysis in the previous part of the article indicates that the regional characteristics of high-quality development of China's insurance industry are characterized by significant differences between the two poles. Provinces with high development levels and those with low development levels coexist, making it more suitable to group them based on larger interquartile points.

Thirdly, there are significant differences between provinces with higher levels of development, especially in the eastern region. The non-parametric kernel density map shows a lack of "peaks", indicating that the development level is relatively scattered. However, even provinces with lower levels of development in the eastern region are better than those with higher levels of development in the central and western regions. For details, please refer to the Dagum Gini coefficient, where the contribution of super variable density is decreasing year by year. Therefore, if the number of grouping boundaries between the third and fourth groups selected is too low, it will not be possible to present the dynamic changes of regions with higher levels of development. For example, in 2016, the 75th percentile was 0.4951, which is higher than the overall 75th percentile of 0.4829. However, in 2016, both the 25th and 50th percentiles were too high, so the dynamic changes in areas with low levels of high-quality development in the insurance industry cannot be presented.

Based on the above considerations, we ultimately chose to group based on the quartiles of the overall year.

Table 1: Overall Year and Representative Year Quantiles of the Insurance Industry High Quality Development Index

overall year 2014 2015 2016

25%quantile 0.3871 0.3649 0.4140 0.4346

50%quantile 0.4363 0.3965 0.4396 0.4616

75%quantile 0.4829 0.4350 0.4691 0.4951

Table 2 Traditional Markov Matrix and Spatial Markov Matrix Dividing Quartile Points Based on 2015

Spatial lag type t/t+1 Ⅰ Ⅱ Ⅲ Ⅳ Frequency

No-lag Ⅰ 0.71875 0.148438 0.09375 0.039063 128

Ⅱ 0.254902 0.254902 0.313725 0.176471 51

Ⅲ 0.075472 0.207547 0.301887 0.415094 53

Ⅳ 0 0.012821 0.089744 0.897436 78

Ⅰ Ⅰ 0.808511 0.117021 0.031915 0.042553 94

Ⅱ 0.368421 0.263158 0.263158 0.105263 19

Ⅲ 0.375 0.25 0.25 0.125 8

Ⅳ 0 0.111111 0 0.888889 9

Ⅱ Ⅰ 0.529412 0.235294 0.235294 0 17

Ⅱ 0.363636 0.090909 0.181818 0.363636 11

Ⅲ 0 0.333333 0.5 0.166667 6

Ⅳ 0 0 0.166667 0.833333 6

Ⅲ Ⅰ 0.466667 0.133333 0.333333 0.066667 15

Ⅱ 0.083333 0.333333 0.583333 0 12

Ⅲ 0 0.2 0.36 0.44 25

Ⅳ 0 0 0.25 0.75 16

Ⅳ Ⅰ 0 1 0 0 2

Ⅱ 0.111111 0.333333 0.222222 0.333333 9

Ⅲ 0.071429 0.142857 0.142857 0.642857 14

Ⅳ 0 0 0.042553 0.957447 47

Table 3 Traditional Markov Matrix and Spatial Markov Matrix for Dividing Quartile Points Based on overall year

Spatial lag type t/t+1 Ⅰ Ⅱ Ⅲ Ⅳ Frequency

No-lag Ⅰ 0.635294 0.329412 0.035294 0 85

Ⅱ 0.197674 0.348837 0.383721 0.069767 86

Ⅲ 0.035714 0.154762 0.452381 0.357143 84

Ⅳ 0 0 0.127273 0.872727 55

Ⅰ Ⅰ 0.613636 0.386364 0 0 44

Ⅱ 0.259259 0.444444 0.259259 0.037037 27

Ⅲ 0 0.666667 0.333333 0 3

Ⅳ 0 0 0.25 0.75 4

Ⅱ Ⅰ 0.741935 0.225806 0.032258 0 31

Ⅱ 0.322581 0.258065 0.354839 0.064516 31

Ⅲ 0.105263 0.157895 0.578947 0.157895 19

Ⅳ 0 0 0.25 0.75 4

Ⅲ Ⅰ 0.444444 0.333333 0.222222 0 9

Ⅱ 0 0.380952 0.52381 0.095238 21

Ⅲ 0 0.152174 0.5 0.347826 46

Ⅳ 0 0 0.142857 0.857143 21

Ⅳ Ⅰ 0 1 0 0 1

Ⅱ 0 0.285714 0.571429 0.142857 7

Ⅲ 0.0625 0.0625 0.1875 0.6875 16

Ⅳ 0 0 0.076923 0.923077 26

5.Consider extending the analysis in Section 4.4.2 with local Moran's index for a more comprehensive examination.

Thank you for your suggestion. We have conducted the local Moran's I test as per your suggestions and included the Moran scatter plots in Figures 8, 9, and 10 of the Figure file. During this revision process, we also re-conducted the global Moran's I test and found that the "internal-external structural coordination" dimension exhibits significant spatial autocorrelation, while the results for the "inclusiveness" and "sustainability" dimensions remain unchanged. These results have been incorporated into the revised manuscript.

Subsequently, we conducted a local Moran index test using data from the year 2020. The Moran scatter plots of the three dimensions showed no significant difference from the global Moran index results. The "inclusiveness" and "internal external structural coordination" dimension indices of most provinces in China were in the first and third quadrants, indicating a significant positive correlation.

6.Incorporate a discussion section to provide insights and interpretations of the results.

Thank you for your suggestion. Based on the analysis in the Results section, we systematically examined the temporal distribution, spatial distribution, dynamic transitions, and regional correlations of high-quality development in China's insurance industry. Next, we employ the Dagum Gini coefficient to further discuss and analyze the changes in spatial disparities and the main factors contributing to these differences in the high-quality development of China's insurance industry. Our findings reveal that the primary factor driving these disparities is regional differences, with inter-regional disparities being more significant than intra-provincial disparities, leading to a gradually evident gradient effect.

7.Verify the format of figures depicting China's map and correct any discrepancies. Additionally, ensure accuracy in references and include international paper references.

Thank you for your suggestion. Due to the graphic format and copyright issues of Chinese maps, we have decided to delete map and submit only the "Basic Parameters of Standard Deviation Ellipse" table（Table 2 in the manuscript）. This table quantifies the results of the standard deviation ellipse method. While it is not as visually intuitive as directly displaying the map, the same conclusions can be drawn based on the specific data changes presented. If readers require the map results, we can provide them upon request.

Additionally, we have revised the format of the literature review and added international paper. We have also added the DOIs of the references and the websites of the literature sources at the end of the references to ensure the accuracy and completeness of the literature review format.

8.Table 7: Calculation Results of Dagum Gini coefficient. One of the year data in the Dagum Gini coefficient composition is incomplete. Please confirm

Thank you for your suggestion.. We have carefully reviewed the results of the Dagum Gini coefficient decomposition and have added the missing 2015 section. Additionally, we have attached the Matlab code used for the Dagum Gini coefficient decomposition.

To ensure that the input data matches the spatial weight matrix, we used the "High Quality" data. This means that after calculating the comprehensive index results for each province, only the comprehensive index was retained. We then converted the long panel data into "wide panel data" and arranged it according to the order of the spatial weight matrix regions.

Response to Reviewer 2 Comments

(1).Introduction:

Further refine the marginal contribution of the article. There have been some studies on the measurement of the level of high-quality development of the insurance industry in the literature. How does this article differentiate from existing literature? The article also mentions " Currently, there is a lack of research on the construction of a comprehensive evaluation system for high-quality development in the insurance industry." Where does this point reflect? It is recommended that the author further elaborate and explain.

Thank you for your suggestion. In response, we have added detailed supplements to the last paragraph of the Introduction section regarding the marginal contributions of this paper. We have integrated the existing literature on measuring the high-quality development level of the insurance industry and further elaborated on the marginal contributions of this paper from various perspectives, including the starting point of measuring the high-quality development level of the insurance industry, theoretical interpretation of the indicators, and the considerations involved.

At present, research on the high-quality development of China's insurance industry is relatively limited, primarily characterized by the following aspects:

Lack of Industry-Level Studies: There is a scarcity of research at the industry level concerning the high-quality development of China's insurance industry. Existing studies predominantly focus on specific sectors within the insurance industry, such as agricultural insurance, health insurance, and life insurance. These studies often investigate the high-quality development of individual insurance products, leaving a gap in research that comprehensively examines the high-quality development of the entire insurance industry. For instance, Yuan Chunqing (2023), Yu Tianyi et al. (2024), Wang Wan (2023), Jia Hongbo et al. (2023), and Xu Xian et al. (2023) have conducted research in various dimensions of insurance, but there is limited examination from the perspective of the overall insurance industry.

Theoretical Deficiency: There is a lack of theoretical research concerning the high-quality development of the insurance industry. Existing studies tend to provide descriptive explanations without adequate grounding in relevant economic theories. There is a notable absence of theoretical frameworks elucidating the relationship between the high-quality development of the insurance industry and its historical development trajectory. Thus, this study seeks to address this gap by exploring theoretical frameworks derived from "quality" and "development" theories to underpin discussions on high-quality development within the insurance industry.

Insufficient Quantitative Research: Quantitative research on the high-quality development of the insurance industry is limited. While some studies, such as those by Hou Xuhua and Tang Yuhui (2022) and Jia Hongbo et al. (2023), have conducted quantitative explorations within specific segments of the insurance industry, there is a dearth of comprehensive quantitative analyses covering the entire insurance sector. This paper aims to contribute to this area of research by conducting quantitative investigations encompassing the entirety of the insurance industry.

In summary, there is a need for more comprehensive and theoretically grounded research at the industry level that utilizes quantitative methodologies to explore the high-quality development of Ch

---

## [Decision Letter · Decision Letter 1]

28 Aug 2024

PONE-D-24-01860R1Spatio-temporal Evolution and Distribution Characteristics of the High-Quality Development of China's Insurance IndustryPLOS ONE

Dear Dr. ZHU,

Thank you for submitting your manuscript to PLOS ONE. After careful consideration, we feel that it has merit but does not fully meet PLOS ONE’s publication criteria as it currently stands. Therefore, we invite you to submit a revised version of the manuscript that addresses the points raised during the review process.

We look forward to receiving your revised manuscript.

Kind regards,

Yu kun Wang

Academic Editor

PLOS ONE

Reviewers' comments:

Reviewer's Responses to Questions

**Comments to the Author**

1. If the authors have adequately addressed your comments raised in a previous round of review and you feel that this manuscript is now acceptable for publication, you may indicate that here to bypass the “Comments to the Author” section, enter your conflict of interest statement in the “Confidential to Editor” section, and submit your "Accept" recommendation.

Reviewer #1: (No Response)

2. Is the manuscript technically sound, and do the data support the conclusions?

Reviewer #1: Partly

3. Has the statistical analysis been performed appropriately and rigorously? 

Reviewer #1: Yes

4. Have the authors made all data underlying the findings in their manuscript fully available?

Reviewer #1: Yes

5. Is the manuscript presented in an intelligible fashion and written in standard English?

Reviewer #1: Yes

6. Review Comments to the Author

Reviewer #1: Thank you for submitting your manuscript. Your work holds significant value and innovation in exploring the high-quality development of the insurance industry. I have carefully reviewed your research and provide the following suggestions to help you further refine your work and enhance its academic contribution and practical value.

Improvement Suggestions:

Integration of Dimensions with Practices:

Although the dimensions of "Inclusiveness," "Sustainability," and "Internal-External Structural Coordination" proposed in the paper are valuable, further clarification is needed on how these dimensions are integrated with specific practices in the insurance industry and their manifestations in different insurance products. It is recommended that you provide concrete examples or empirical data to support the practical application of these dimensions.

Enhancement of the Indicator System:

It is recommended to include an indicator of insurance consumer satisfaction in the index system to more comprehensively assess the high-quality development of the insurance industry. Additionally, each indicator’s selection should be supported by a more robust theoretical basis and empirical evidence. Please add relevant literature support and empirical data to justify the choice of each indicator.

Transparency of Data Sources:

The paper mentions the removal of map images due to copyright issues. It is suggested that you provide more information on data sources, including the timing of data acquisition, processing methods, and a declaration of dataset completeness to enhance the transparency of the research.

Explanation of Analytical Methods:

While the paper employs various analytical methods, the rationale for selecting each method and the conditions for their applicability are not sufficiently explained. Particularly for the Spatial Markov Chain model, it is advised to provide a detailed description of the selection process for model parameters and the interpretation of model results.

Analysis of Regional Differences:

The results section shows the development levels of the insurance industry in different regions, but the analysis of the causes of these differences is insufficient. It is advised to explore the reasons for inter-regional differences more deeply by considering factors such as regional economy, policy environment, and market demand.

Improvement of the Literature Review:

The literature review section should more precisely identify the shortcomings of existing research and the innovations of this study, avoiding vague descriptions to enhance the clarity and relevance of the literature review.

Discussion and Conclusion:

The discussion section should delve deeper into the implications of the research findings and explore their implications for policy-making and practice in the insurance industry. The conclusion section should more clearly summarize the main findings of the study, highlight its limitations, and suggest directions for future research.

Relevance of Supplementary Materials:

The supplementary materials mentioned in the paper should ensure they are closely related to the content of the main text and easy for readers to understand, avoiding unrelated or redundant information.

Specific Policy Recommendations:

The paper should provide more specific policy recommendations to help decision-makers and practitioners understand how to promote the high-quality development of the insurance industry.

7. PLOS authors have the option to publish the peer review history of their article (what does this mean? ). If published, this will include your full peer review and any attached files.

**Do you want your identity to be public for this peer review?** For information about this choice, including consent withdrawal, please see our Privacy Policy .

Reviewer #1: No

---

## [Author Response · Author response to Decision Letter 1]

4 Sep 2024

Response to Reviewer 1 Comments

For this article, I suggest that the author correct it as follows

(a) The derivation of equations (1) to (9) does not seem to be connected to the content

of Table 3 in the article.

Thank you for the reviewer’s suggestions. Formulas (1) through (9) outline the calculation process for the comprehensive index. The results of formulas (1) through (5), which represent the weight coefficients of each sub-indicator, are presented in Table 1. The outcomes of formulas (6) through (9) are included in the High-Quality.dta file. The results displayed in Table 3 were derived using the standard deviation ellipse calculation program available in ArcGIS software, by inputting the results from formulas (1) through (9) into the program. The calculation formula for the basic parameters of the ellipse, as presented in Table 3, is as follows. We have also added these formulas to the article. (see lines 583 to 587 of the revised manuscript):

(b) In addition, the empirical results in Table 3 do not include the variable self-

attribution test, the single root test, the co-integration test, and other pre-statistical tests, causality tests, etc.

Thank you for the reviewer's suggestions. The standard deviation ellipse method presented in Table 3 is not a regression technique within the field of econometrics. Instead, it is used to analyze the spatial distribution of high-quality development in the insurance industry across the country. As such, there is no relevant test for this method as you mentioned.

(c)In addition, please confirm whether the 2011 and 2012 years of Inclusiveness in

Table 3 should be 2015 and 2020

Thank you for the reviewer’s correction. We sincerely apologize for this oversight, and we will make the necessary revisions to the relevant sections. We appreciate the reviewer's thorough and conscientious review.

(d) There are too few references and it is not detailed whether the formula was created by the author himself or a reference, and the author is asked to explain why formulas (1) to (9) are used, and whether the model has any shortcomings, rather than other more suitable models to analyze

Thank you for the reviewers' suggestions. Currently, empowerment methods are categorized into subjective and objective approaches. Subjective methods include techniques such as expert scoring and the analytic hierarchy process, while objective methods encompass the vertical and horizontal grading method, the CEITIC method, and the entropy weight method. It is crucial to balance the objectivity and the inherent information content of the data. In our research on the spatial distribution of the high-quality development index, we have employed the entropy weight TOPSIS method, as supported by previous studies (Yang[1], Nie[2], Liu[3], etc.). We have also included the relevant references in our article.

we have also expanded the explanations of each model in the text and included the formulas used for calculating the results.

The kernel density estimation method: Lines 475 to 491.

The Standard Deviation Ellipse (SDE) method：Lines 580 to 591.

The traditional Markov chain：Lines 617 to 626.

The spatial Markov chain：Lines 631 to 643.

Moran's Index：Lines 733 to 742.

Response to Reviewer 2 Comments

(1) Although the dimensions of "Inclusiveness," "Sustainability," and "Internal-External Structural Coordination" proposed in the paper are valuable, further clarification is needed on how these dimensions are integrated with specific practices in the insurance industry and their manifestations in different insurance products. It is recommended that you provide concrete examples or empirical data to support the practical application of these dimensions.

Thank you for the reviewer's suggestions. In response, we have expanded the section titled "Theoretical Connotations of High-Quality Development of the Insurance Industry" (see lines 206 to 261 of the revised manuscript) as per your recommendation. To better illustrate how these concepts align with the current realities of China's insurance industry, we have included relevant use cases and data.

(2) Enhancement of the Indicator System:

Thank you for the reviewer's suggestions. It is recommended to include an indicator of insurance consumer satisfaction in the index system to more comprehensively assess the high-quality development of the insurance industry. Additionally, each indicator’s selection should be supported by a more robust theoretical basis and empirical evidence. Please add relevant literature support and empirical data to justify the choice of each indicator.

Thank you for the reviewer's suggestions. We considered including customer satisfaction in our indicator system. However, due to the practical limitations, we have focused on constructing a macro-level indicator system for the high-quality development of the insurance industry. Customer satisfaction, being a micro-level indicator, is not compatible with this approach and is also unavailable at the municipal level. In the future, we plan to develop a micro-level indicator system for the high-quality development of the insurance industry, where we will incorporate customer satisfaction as an indicator based on your feedback. In addition, due to the duplication of references for many indicators, and considering the length, only the dimensional indicators are listed as references, while the calculation of the tertiary indicators is reflected in Table 1. We hope the reviewers can understand. The specific content of the selection basis for indicators in the article is as follows (see lines 293 to 314 of the revised manuscript):

(3)Transparency of Data Sources:

Thank you for the reviewer's suggestions. The details of the data sources in the article are as follows（see lines 354 to 378 of the revised manuscript）:

The main data sources include the "Statistical Yearbooks" of various provinces in China from 2010 to 2020, the "Insurance Statistical Yearbook," and the websites of the financial regulatory bureaus of each province. Some data require processing, and the details are as follows: (1) Due to differences in the statistical scope of insurance institutions among provinces in terms of time and space, rendering them incomparable, the number of insurance companies at the provincial level and above is selected as the indicator. The data on the number of insurance companies in each province are sourced from the provincial "Financial Operation Reports" and the regional version of the "Insurance Statistical Yearbook."(2) Data on the number of insurance industry employees are sourced from the Chinese Labor Economic Database in the EPS database, specifically the number of urban insurance industry employees at the end of the year. (3) For some provinces and years with missing insurance amount data, linear interpolation after logarithmic transformation of insurance amounts is conducted to complete the data. (4) Due to the lack of data on the number of legal entities in each province before 2013, the number of industrial enterprises in each province is selected as a proxy indicator based on data availability and comparability. The data are sourced from the Wind database. (5) Data on the number of enterprises held by insurance institutions in each province are sourced from the China National Research Data Sharing Platform (CNRDS), specifically from databases on shareholding of listed companies and basic information of listed companies. (6) Long-term trends are calculated using the HP filter method. Since the selected premium income, claims, and benefits paid are annual data, a parameter λ=100 is chosen based on previous scholars' experiences to calculate the long-term trend. The deviation between the data for the current year and the long-term trend is considered as the deviation data. (7) The number of insurance technology companies is obtained by searching relevant keywords such as "insurance technology" and "internet insurance" (8) The HHI (Herfindahl-Hirschman Index) is calculated based on the market share of premium income of each insurance company in each province, as provided in the "Insurance Statistical Yearbook."

(4) The paper mentions the removal of map images due to copyright issues. It is suggested that you provide more information on data sources, including the timing of data acquisition, processing methods, and a declaration of dataset completeness to enhance the transparency of the research.

Thank you for the reviewer's suggestion. We did not display the map due to concerns regarding map copyright. The shp file of the China map used in our study was obtained from the National Geographic Center of China (https://www.ngcc.cn/). After importing the shapefile into ArcGIS software, we generated the basic framework of China's provincial map and then input the relevant data for calculation. Details about the data source, its integrity, and our methodology were explained earlier in the manuscript. The data used to create the chart can be found in the High-Quality.dta file we provided.

(5) Explanation of Analytical Methods:

While the paper employs various analytical methods, the rationale for selecting each method and the conditions for their applicability are not sufficiently explained. Particularly for the Spatial Markov Chain model, it is advised to provide a detailed description of the selection process for model parameters and the interpretation of model results.

Thank you to the reviewers for your comments. In response, we have expanded the explanations of each model in the text and included the formulas used for calculating the results.

The kernel density estimation method:Lines 475 to 491.

The Standard Deviation Ellipse (SDE) method：Lines 580 to 591.

The traditional Markov chain：Lines 617 to 626.

The spatial Markov chain：Lines 631 to 643.

Moran's Index：Lines 733 to 742.

(6)Analysis of Regional Differences:

The results section shows the development levels of the insurance industry in different regions, but the analysis of the causes of these differences is insufficient. It is advised to explore the reasons for inter-regional differences more deeply by considering factors such as regional economy, policy environment, and market demand.

Thank you for the reviewers' comments. In this revision, we have added an analysis of potential reasons for each result, taking into account factors such as economic development level, economic structure, and regional policies.(see lines 512 to 516; Lines 539 to 543; Lines 663 to 667; Lines 750 to 757)

(7) Improvement of the Literature Review:

The literature review section should more precisely identify the shortcomings of existing research and the innovations of this study, avoiding vague descriptions to enhance the clarity and relevance of the literature review.

Thank you to the reviewers for their comments. In this revision, we have expanded the "Introduction" section (see lines 61 to 95 of the revised manuscript) to include a discussion of the existing literature on measuring high-quality development and the development of the insurance industry. We also identified the shortcomings in the current research and explained how our study addresses these gaps.

(8) Discussion and Conclusion:

The discussion section should delve deeper into the implications of the research findings and explore their implications for policy-making and practice in the insurance industry. The conclusion section should more clearly summarize the main findings of the study, highlight its limitations, and suggest directions for future research.

Thank you to the reviewers for their comments. In the conclusion section, we have elaborated on the research directions, identified research deficiencies, and outlined plans for further research. This addition aims to clarify our research positioning and future research agenda.(see lines 928 to 932 of the revised manuscript)

(9) Relevance of Supplementary Materials:

The supplementary materials mentioned in the paper should ensure they are closely related to the content of the main text and easy for readers to understand, avoiding unrelated or redundant information.

Thank you to the reviewer for your suggestions. In this revision, we have thoroughly reviewed the supplementary materials to enhance the readability of our research paper.

(10) Specific Policy Recommendations:

The paper should provide more specific policy recommendations to help decision-makers and practitioners understand how to promote the high-quality development of the insurance industry.

Thank you to the reviewer for your suggestions. We have enhanced the policy recommendations by including specific examples and suggestions from the perspective of insurance companies, thereby making our recommendations more concrete and actionable(see lines 895 to 898; lines 911 to 914).

---

## [Editor Report · Decision Letter 2]

12 Sep 2024

PONE-D-24-01860R2Spatio-temporal Evolution and Distribution Characteristics of the High-Quality Development of China's Insurance IndustryPLOS ONE

Dear Dr. ZHU,

Thank you for submitting your manuscript to PLOS ONE. After careful consideration, we feel that it has merit but does not fully meet PLOS ONE’s publication criteria as it currently stands. Therefore, we invite you to submit a revised version of the manuscript that addresses the points raised during the review process.

We look forward to receiving your revised manuscript.

Kind regards,

Yu kun Wang

Academic Editor

PLOS ONE
---

## [Author Response · Author response to Decision Letter 2]

13 Sep 2024

Response to Academic Editor Comments

1.I would like to ask: The numerator of formula 20 has two identical i=1 to N, which seems to be wrong Please explain

Thank you very much for the editor's comments. Due to an error, we mistakenly included a summation symbol when writing the formula. The correct form of formula 20 should be as follows:

S^2=i=1nx_i-x2n#20

The calculation of S^2 is based on sample variance. We have made the necessary correction on line 740 (Formula 20) of the revised manuscript.

In addition, we sincerely apologize for this mistake. We have reviewed and checked the formulas throughout the text to ensure their accuracy. We apologize once again and greatly appreciate your understanding.

Response to Journal Requirements:

Thank you very much for the editor's comments. Upon reviewing the citations in the full text, we found that some references were inadvertently omitted during the revision process. In this revision, we have supplemented the missing references and indicated their citation positions in the text (see lines 67 and 543 of the revised manuscript). We have also rechecked the reference format and made corresponding adjustments.

Thank you very much for the editor's comments. We have upload our figure files to the Preflight Analysis and Conversion Engine (PACE) digital diagnostic tool, and ensure that figures meet PLOS requirements. We will place the latest images in the figure. word file and upload to the system.

---

## [Editor Report · Decision Letter 3]

23 Sep 2024

Spatio-temporal Evolution and Distribution Characteristics of the High-Quality Development of China's Insurance Industry

PONE-D-24-01860R3

Dear Dr. ZHU,

We’re pleased to inform you that your manuscript has been judged scientifically suitable for publication and will be formally accepted for publication once it meets all outstanding technical requirements.

Kind regards,

Yu kun Wang

Academic Editor

PLOS ONE
---

## [Editor Report · Acceptance letter]

PONE-D-24-01860R3

PLOS ONE

Dear Dr. ZHU,

I'm pleased to inform you that your manuscript has been deemed suitable for publication in PLOS ONE. Congratulations! Your manuscript is now being handed over to our production team.

Kind regards,

on behalf of

Professor Yu kun Wang

Academic Editor

PLOS ONE